# Predictor-Corrector Enhanced Transformers with Exponential Moving Average Coefficient Learning

**Bei Li**[1,2*]   **Tong Zheng**[1]   **Rui Wang**[3]   **Jiahao Liu**[2]   **Qingyan Guo**[4]   **Junliang Guo**[3]
**Xu Tan**[3]   **Tong Xiao**[1†]   **Jingbo Zhu**[1]   **Jingang Wang**[2†]   **Xunliang Cai**[2]
[1]Northeastern University   [2]Meituan Inc.   [3]Microsoft Research   [4]Tsinghua University
{libei17,liujiahao12,wangjingang02,caixunliang}@meituan.com
tzheng24@umd.edu   {ruiwa,junliangguo,xuta}@microsoft.com
gqy22@mails.tsinghua.edu.cn   {xiaotong,zhujingbo}@mail.neu.edu.cn

## Abstract

Residual networks, as discrete approximations of Ordinary Differential Equations (ODEs), have inspired significant advancements in neural network design, including multistep methods, high-order methods, and multi-particle dynamical systems. The precision of the solution to ODEs significantly affects parameter optimization, thereby impacting model performance. In this work, we present a series of advanced explorations of Transformer architecture design to minimize the error compared to the true "solution." First, we introduce a predictor-corrector learning framework to minimize truncation errors, which consists of a high-order predictor and a multistep corrector. Second, we propose an exponential moving average-based coefficient learning method to strengthen our higher-order predictor. Extensive experiments on large-scale machine translation, abstractive summarization, language modeling, and natural language understanding benchmarks demonstrate the superiority of our approach. On the WMT'14 English-German and English-French tasks, our model achieved BLEU scores of 30.95 and 44.27, respectively. Furthermore, on the OPUS multilingual machine translation task, our model surpasses a robust 3.8B DeepNet by an average of 2.9 SacreBLEU, using only 1/3 parameters. Notably, it also beats LLama models by 5.7 accuracy points on the LM Harness Evaluation.

## 1   Introduction

Residual networks [16], formally $y_{t+1} = y_t + \mathcal{F}(y_t, \theta_t)$, represent a cornerstone in the development of deep neural networks [59, 10], primarily due to their capacity to facilitate the flow of information across multiple layers. Beyond their pivotal role in convolutional networks, residual connections have become an essential element in the architecture of more complex models, including the Transformer [59] and its various derivatives. This concept can be likened to the discretization process in the Euler method [63, 37, 15, 7, 51, 29], which serves as a first-order solver for ordinary differential equations (ODEs), where $\frac{dy(t)}{dt} = \mathcal{F}(y(t), \theta(t))$. In both cases, the new state (be it the next layer's output in ResNets or the solution at the next time step in the Euler method) is computed by taking the current state and adding an adjustment term.

Given this analogy with ODEs, there has been a surge of interest in improving residual network architectures by using more powerful numerical methods for ODEs. For instance, the linear multistep method [62, 37, 71, 24] has been employed to bolster the optimization of deep models. Other efforts have included redesigning the Transformer architecture from a multi-particle dynamical system

---

*Part of work was done during an internship at Microsoft Research Asia
†Corresponding Author.

38th Conference on Neural Information Processing Systems (NeurIPS 2024).

perspective [38, 12] and improving parameter learning efficiency through high-order methods [29]. Additionally, ODEs have been extensively studied for their potential to accelerate diffusion processes, with multistep and high-order solvers offering more accurate predicted noise among each denoising process, generating comparable images but consuming much fewer NFEs [31, 35, 36].

In this work, we mainly focus on advancing the architecture design, specifically by minimizing the truncation error across each timestep. Building upon the ODE Transformers [29], which replace the first-order Euler method with a high-order method for more precise numerical solutions, our focus extends to addressing two key limitations. First, high-order solutions, such as those from the Runge-Kutta method or multistep methods, are found not to lead to significant improvements when we scale up training data and/or model size. Second, the gated fusion coefficient learning method, which is widely used in previous work, is not well suited for higher-order solutions.

Our work draws inspiration from the predictor-corrector method [11], a well-established approach in numerical analysis known for its accuracy in solving differential equations. This method involves a two-step process: a prediction step that estimates the solution based on known conditions, followed by a correction step that refines the prediction for a more accurate result. We introduce a novel family of PCformers that embrace this predictor-corrector paradigm. Our approach integrates the final solution using an exponential moving average (EMA) method, capitalizing on the insight that *higher-order intermediate approximations tend to be more accurate*. This assertion is supported by the truncation error analysis presented in Section 3.1.2. Our method is not only readily extensible to arbitrary higher orders but also consistently outperforms the gated fusion method.

Our contributions are summarized below:

- We extend explicit ODE solutions to implicit ODE solutions via a predictor-corrector learning paradigm. This kind of iterative refinement can attain more accurate solutions than previous studies both theoretically and empirically. In particular, we choose the high-order method as the predictor and the multistep method as the corrector.

- To further strengthen the learning ability and training stability for high-order methods, we propose an exponential moving average coefficient learning method to replace the constant coefficients. This leads to a much stronger predictor.

- Our extensive experimental evaluation on several benchmarks, including WMT'14 English-German, WMT'14 English-French, WMT'16 Romanian-English, and the OPUS multilingual machine translation benchmark, demonstrates the superior effectiveness of our PCformer models. Notably, our model surpasses the 3.8B DeepNet by an average BLEU score of 2.9 with only 1/3 of the parameters. Furthermore, our model can be extended to other domains. Results on abstractive summarization, language modeling, and language understanding tasks demonstrate its generality.

## 2   Background

We build our method upon Transformer [59] as it is one of the most popular models in NLP. The encoder is a stack of identical layers. Each layer consists of a self-attention block and a feedforward network (FFN) block. Both of them are equipped with a residual connection [16] and a layer normalization unit [25]. The output of a block can be defined as

$$y_{t+1} = y_t + \mathcal{F}(y_t, \theta_t) \tag{1}$$

where $\mathcal{F}(\cdot)$ is either the self-attention or FFN block. This equation illustrates that the layer output $y_{t+1}$ is determined by the layer input $y_t$ and a learnable derivative estimated by the current function $\mathcal{F}$. This approach aligns with the benefits of the Euler method, which we will discuss in the following sections. In this work, we use $F_t$ to denote the $t$-th layer representation and $\hat{F}_i$ to denote the $i$-th order intermediate approximations.

**The Euler Method**   The Euler method is the most basic solution to solve ODEs given the initial value, involving a function $y(t)$ of a variable $t$ and its derivatives. The Euler method defines the first-order derivative of $y(t)$

$$\frac{\mathrm{d}y(t)}{\mathrm{d}t} = f(y(t), t) \tag{2}$$

where $f(y(t), t)$ defines a time-dependent vector field if we know its value at all points of $y$ and all instants of time $t$. Eq. 2 illustrates that the change of a variable is determined by its current value

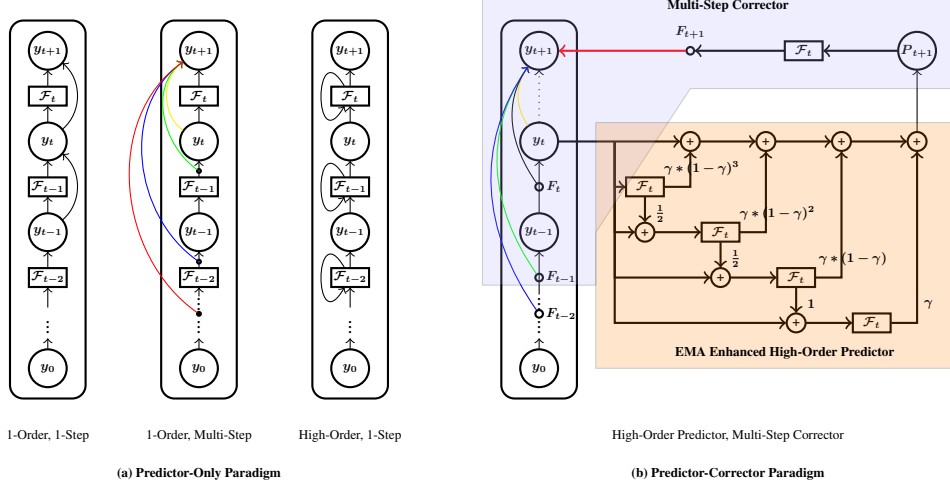

Figure 1: Illustration of several advanced numerical methods and our proposed predictor-corrector paradigm. The right part plots a 4-order method as the predictor to obtain $P_{t+1}$; $F_{t+1}$ is then estimated via a function $\mathcal{F}(\cdot)$; A 4-step method as the corrector to obtain the $y_{t+1}$.

and a time variable $t$. In deep learning, we can use a trainable function $\mathcal{F}(\cdot)$ to estimate $f(y(t), t)$. In a nutshell, residual networks could be regarded as a 1st-order discretization of the Euler method [63, 29]. The advantage is obvious since residual networks deliver consistent performance gains in the artificial intelligence, but the precision of $y_{t+1}$ is limited. Fortunately, we can move forward along the numerical analysis perspective as more advanced numerical methods can alleviate this issue.

**The Linear Multistep Method** Compared with the Euler method, the linear multistep method uses previously obtained "solutions" to estimate the current one, leading to more accurate results. Formally, a multistep method could be defined as: $y_{t+1} = y_t + \sum_{i=1}^{t} \alpha_i F_i$, where $F_t = \mathcal{F}(y_t, \theta_t)$.

**The High-Order ODE Method** Another family of numerical methods is high-order ODE solvers by repeatedly refining the solutions within a single step. Previous work [29] employed the Runge-Kutta methods [50, 23, 6, 2] for a higher-order solution to ODEs, where Runge-Kutta is a classic family of iterative methods with different orders of precision. More formally, the explicit Runge-Kutta methods of an $n$-order solution is defined to be: $y_{t+1} = y_t + \sum_{i=1}^{n} \gamma_i \hat{F}_i$, where $\hat{F}_1 = \mathcal{F}(y_t, \theta_t)$, $\hat{F}_i = \mathcal{F}(y_t + \sum_{j=1}^{i-1} \beta_{ij} \hat{F}_j, \theta_t)$. Note that $\hat{F}_i$ is the intermediate approximation to the solution at an inner step. $\beta$ and $\gamma$ are coefficients to model the scale of the input and the output of $\hat{F}_i$. This kind of method can be adapted to Transformer blocks by reusing $\mathcal{F}(\cdot)$ within a block.

# 3 Predictor-Corrector Transformer

In this section, we first show the core design of Predictor-Corrector paradigm to more accurately solve ODEs. Then we propose an alternative coefficient learning strategy that could be applied to arbitrary orders using the merit of the exponential moving average. At last, we show some additional training techniques for stable and well-performance training.

## 3.1 Predictor-Corrector Method

> *A genuine problem-solving process involves the repeated use of available information to initiate exploration, which discloses, in turn, more information until a way to attain the solution is finally discovered. – Newell et al. [43]*

The Predictor-Corrector framework leverages an iterative process of using available information to refine approximations continuously. Initially, the Predictor generates a rough estimate, which is subsequently refined by the Corrector using newly available data. This cyclical process mirrors the

problem-solving strategy described earlier, where each iteration uncovers additional information that enhances the final solution's accuracy.

### 3.1.1 Adams-Bashforth-Moulton Methods

A predictor-corrector method typically uses an explicit method for the predictor and an implicit method for the corrector. Here we take the 4-step Adams-Bashforth-Moulton [1] methods as an instance, where Adams-Bashforth is the predictor and Adams-Moulton is the corrector. Adams-Bashforth is a 4-step method, which defined as

$$y_{t+1} = y_t + \frac{1}{24}(55F_t - 59F_{t-1} + 37F_{t-2} - 9F_{t-3}), \tag{3}$$

where $F_t = \mathcal{F}(y_t, \theta_t)$. $\mathcal{F}(\cdot)$ denotes the $t$-th function, and $\theta_t$ is corresponding parameters. Obviously, the Adams-Bashfroth methods are explicit since $y_{t+1}$ only depends on the "observed" statistics ($F_{i \leq t}$). Similarly, Adams-Moulton is also a 4-step method as below

$$y_{t+1} = y_t + \frac{1}{24}(9F_{t+1} + 19F_t - 5F_{t-1} + F_{t-2}). \tag{4}$$

Formally, both Eq. 3 and Eq. 4 reused $F_t$, $F_{t-1}$, $F_{t-2}$ to improve the accuracy, but the corrector necessitates an approximate current "solution" $F_{t+1}$ to substitute $F_{t-3}$, which is an implicit method. This is because that $y_{t+1}$ is the value to be solved, thus we cannot compute $F_{t+1}$. To solve this, the Adams-Bashforth-Moulton methods utilize Eq. 3 to obtain the approximate value ($P_{t+1}$) for $y_{t+1}$. Then $F_{t+1}$ could be approximated following $F_{t+1} = \mathcal{F}(P_{t+1}, \theta_t)$. Concretely, the predictor provides a rough approximation, which is the combination of the preceding four layer representations. And the corrector then improves the approximation, offering a more precise sample derived from the data.

However, applying the Adams-Bashforth-Moulton method directly to Transformer architecture design leads to unstable training and limited benefits due to the difficulty in optimizing constant coefficients. Similar issues have been observed in training a Runge-Kutta (RK4) network with numerically suggested coefficients [29]. To address these challenges, Wang et al. [62] proposed a Dynamic Linear Combination of Layers (DLCL) method. This approach utilizes learnable coefficients and adjusts steps based on layer depth, effectively transforming it into a variable multistep method. Additionally, the Adams-Bashforth method is not the only choice for the predictor; other numerical methods, such as high-order methods, are also considered strong alternatives as they often provide more accurate solutions [29, 72].

### 3.1.2 High-order Predictor and Multistep Corrector

The aforementioned discussion motivates us to design a more powerful and stable architecture based on the above principles. Our preliminary experiments indicate that more accurate predictors indeed improve the performance, thus we choose a high-order method to serve as the predictor. A 2-order method could be defined as

$$y_{t+1} = y_t + \frac{1}{2}(\hat{F}_1 + \hat{F}_2) \tag{5}$$

where $\hat{F}_1 = \mathcal{F}(y_t, \theta_t)$, and $\hat{F}_2 = \mathcal{F}(y_t + \hat{F}_1, \theta_t)$. Rather than utilizing the previously obtained representations in multistep methods, high-order methods iteratively estimate the approximations upon the last timestep. Similarly, a 4-order method is:

$$y_{t+1} = y_t + \frac{1}{6}(\hat{F}_1 + 2\hat{F}_2 + 2\hat{F}_3 + \hat{F}_4) \tag{6}$$

where $\hat{F}_1 = \mathcal{F}(y_t, \theta_t)$, $\hat{F}_2 = \mathcal{F}(y_t + \frac{1}{2}\hat{F}_1, \theta_t)$, $\hat{F}_3 = \mathcal{F}(y_t + \frac{1}{2}\hat{F}_2, \theta_t)$, and $\hat{F}_4 = \mathcal{F}(y_t + \hat{F}_3, \theta_t)$. To break the limit of constant coefficients, Li et al. [29] employed a gated network to dynamically compute the coefficients of $\hat{F}_1$ and $\hat{F}_2$, however, this method cannot applied to higher-order methods, e.g., RK4. To facilitate higher-order optimization, we design a more flexible coefficient learning method via an exponential moving average strategy.

**Predictor with Exponential Moving Average Coefficient Learning**  The Exponential Moving Average (EMA) method [20] is widely used for estimating time-series data by assigning variable weights to past observations, giving more importance to recent data compared to simple weighted averaging methods. We hypothesize that high-order approximations at each step should have a larger

impact on the final output, as they provide a more accurate initial state than previous ones. To support this claim, we replaced Eq. 6 by $y_{t+1} = y_t + \hat{F}_i$, where $i \in [1, ..., 4]$. We used a single-layer decoder to compute the perplexity (PPL) on the validation set to simulate truncation errors. Our expectation is that *the fewer truncation errors, the larger the coefficient it should own*.

Figure 2 displays the perplexity comparisons. It shows that a 4-th order approximation, used as a replacement for the linear aggregation in Eq. 6, delivers comparable results and outperforms other cases. This observation motivates us to combine the benefits of EMA with the coefficient learning method. EMA is more flexible to the order of ODE solvers, that is could be easily extended to 2 orders, 4 orders, or even larger. Figure 1(b) illustrates the design merit of our proposed PCformer with an EMA coefficient predictor and a parameterization multistep corrector. Here, we use the RK4-block as an example. It is apparent that the original scales have been replaced by

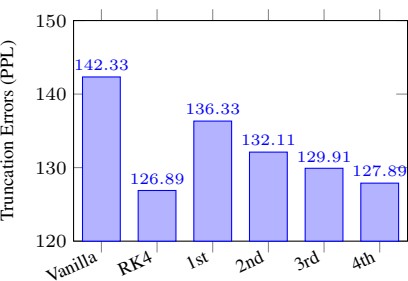

Figure 2: Truncation errors with different intermediate approximations.

$\gamma, \gamma \cdot (1 - \gamma), \gamma \cdot (1 - \gamma)^2$, and $\gamma \cdot (1 - \gamma)^3$ from $\hat{F}_4$ to $\hat{F}_1$, where $\gamma$ is learnable and the initialization is 0.5 empirically. In this way, our $n$-order predictor approximates $P_{t+1}$ as follows:

$$P_{t+1} = y_t + \sum_{i=1}^{n} \gamma \cdot (1 - \gamma)^{n-i} \cdot \hat{F}_i. \tag{7}$$

**Corrector with Parameterization** Leveraging a robust predictor, our corrector is designed to be computationally lightweight, striking an optimal balance between performance and efficiency. Utilizing the Adams-Moulton method, we parameterize the coefficients of previous states with learnable parameters. These coefficients are initialized using an EMA value, where the newly estimated $F_{t+1}$ is assigned a larger weight ($\alpha = 0.5$), and the weights of previous states decrease in a descending order. In this way, we rewrite the Eq. 4 by

$$y_{t+1} = y_t + \alpha \cdot \mathcal{F}(P_{t+1}, \theta_t) + \sum_{i=t-2}^{t} \alpha \cdot (1 - \alpha)^{t-i+1} \cdot F_i. \tag{8}$$

where $P_{t+1}$ is obtained with Eq. 7. Empirically, we found that when the dataset is limited, an Backward Euler method [5] as the corrector is enough to provide precise correction, where $y_{t+1} = y_t + \mathcal{F}(P_{t+1}, \theta_t)$. We will discuss this in the analysis for more insights.

### 3.2 Improving Training Stability

**Step Normalization (RK-Norm)** We built our PCformer following pre-norm architecture [62, 58], by rewritten Eq. 1 to $y_{t+1} = y_t + \mathcal{F}(\text{LN}(y_t), \theta_t)$, where $\text{LN}(\cdot)$ denotes the normalization. This ensures that the representation is normalized before computing the derivative $F_i$. To achieve this, we normalize the obtained intermediate approximations $\hat{F}_i$ at each inner step and then compute the offset, e.g., $y_t + \frac{1}{2}\text{LN}(\hat{F}_1)$ to obtain the $\hat{F}_2$ for the next timestep. Meanwhile, the $\hat{F}_i$ in Eq. 7 is rewritten by $\text{LN}(\hat{F}_i)$. If not, this oversight can cause instability when computing the final ODE solution, where we will make ablations in the analysis section. The algorithm

---

**Algorithm 1** Predictor-Corrector Paradigm

1: **procedure** PREDICTORCORRECTOR($\mathbf{y_t}, \mathbf{H}$)
2:     $\mathbf{S} \leftarrow \emptyset$     ▷ Initialize an empty list to store $\hat{F}_i$
3:     **for** $i \leftarrow 1$ **to** 4 **do**
4:         **if** $i == 1$ **then**
5:             $\hat{F}_1 \leftarrow \mathcal{F}(\mathbf{y_t}, \theta_\mathbf{t})$     ▷ Compute $\hat{F}_1$
6:         **else**
7:             $\hat{\mathbf{F}}_\mathbf{i} \leftarrow \mathcal{F}(\mathbf{y_t}, \mathbf{S[i-1]}, \theta_\mathbf{t})$     ▷ Compute $\hat{F}_i$
8:         **end if**
9:         $\text{LN}(\hat{\mathbf{F}}_\mathbf{i}) \leftarrow \text{LayerNorm}(\hat{\mathbf{F}}_\mathbf{i})$     ▷ Apply RK-Norm
10:         $\mathbf{S}.\text{append}(\text{LN}(\hat{\mathbf{F}}_\mathbf{i}))$     ▷ Store $\text{LN}(\hat{\mathbf{F}}_\mathbf{i})$
11:     **end for**
12:     Compute $P_{t+1}$ using $\mathbf{S}$ via Eq. 7     ▷ Predictor
13:     $\mathbf{F_{t+1}} \leftarrow \mathcal{F}(P_{t+1})$     ▷ Compute $F_{t+1}$
14:     Compute $y_{t+1}$ using $\mathbf{H}$ via Eq. 8     ▷ Corrector
15:     $\mathbf{H}.\text{add}(\mathbf{F_{t+1}})$     ▷ Store $F_{t+1}$
16:     **return** $\mathbf{y_{t+1}}$     ▷ Return the layer output
17: **end procedure**

---

(right part) presents a more detailed computation flow of a single layer in our PCformer, where $\mathbf{H}$ stores the previously obtained $F_{t+1}$.

Table 1: Comparison with the state-of-the-arts on the WMT En-De and WMT En-Fr tasks. We both report the tokenized BLEU and SacreBLEU scores for comparison with previous work.

| Model | Layers | WMT En-De | | | | WMT En-Fr | | | |
|---|---|---|---|---|---|---|---|---|---|
| | | #Param | Steps | BLEU | SBLEU | #Param | Steps | BLEU | SBLEU |
| Transformer [59] | 6-6 | 213M | 100K | 28.40 | - | 222M | 300K | 41.00 | - |
| MacaronNet [38] | 6-6 | - | - | 30.20 | - | - | - | - | - |
| Transformer-DLCL [62] | 30-6 | 137M | 50K | 29.30 | 28.6 | | | | |
| Transformer-Base | 6-6 | 61M | 50K | 27.89 | 26.8 | 69M | 100K | 41.05 | 39.1 |
| RK2-block (Gated) [29] | 6-6 | 61M | 50K | 28.89 | 27.7 | 69M | 100K | 42.31 | 40.3 |
| RK2-block (EMA) | 6-6 | 61M | 50K | 29.11 | 28.1 | 69M | 100K | 42.44 | 40.4 |
| RK4-block [29] | 6-6 | 61M | 50K | 29.03 | 27.9 | 69M | 100K | 42.56 | 40.6 |
| RK4-block (EMA) | 6-6 | 61M | 50K | 29.43 | 28.4 | 69M | 100K | 42.72 | 40.7 |
| Transformer-Big | 6-6 | 211M | 100K | 29.21 | 28.1 | 221M | 100K | 42.89 | 40.9 |
| RK2-block (Gated) [29] | 6-6 | 211M | 100K | 30.53 | 29.4 | 221M | 100K | 43.59 | 41.6 |
| RK4-block [29] | 6-6 | 211M | 100K | 30.39 | 29.3 | 221M | 100K | 43.51 | 41.6 |
| PCformer (2-order) | 6-6 | 211M | 100K | 30.90 | 29.8 | 221M | 100K | 43.85 | 41.8 |
| Transformer-Big | 12-6 | 286M | 100K | 29.91 | 28.9 | 297M | 100K | 43.22 | 41.2 |
| RK2-block (Gated) [29] | 12-6 | 286M | 100K | 30.77 | 29.6 | 297M | 100K | 43.96 | 42.1 |
| RK4-block [29] | 12-6 | 286M | 100K | 30.55 | 29.4 | 297M | 100K | 43.81 | 41.8 |
| RK4-block (EMA) | 12-6 | 286M | 100K | 30.66 | 29.5 | 297M | 100K | 44.17 | 42.2 |
| PCformer (2-order) | 12-6 | 286M | 100K | **30.95** | **29.8** | 297M | 100K | **44.27** | **42.4** |

Table 2: Results on the En-Ro task.

| Model | Params | BLEU |
|---|---|---|
| Transformer in [39] | 62M | 34.30 |
| DeLight [39] | 53M | 34.70 |
| Transformer (Our impl.) | 69M | 33.49 |
| RK2-block (gated) [29] | 69M | 34.94 |
| PCformer (2-order) | 69M | 35.43 |
| PCformer (4-order) | 69M | 35.49 |
| RK4-block [29] | 226M | 35.28 |
| PCformer (2-order) | 226M | 35.55 |
| PCformer (4-order) | 226M | **35.80** |

Table 3: Average SacreBLEU on the OPUS-100.

| Models | Layers | Hidden | Params | X→En | En→X | Avg |
|---|---|---|---|---|---|---|
| DeepNet [61] | 200 | 512 | 863M | 33.2 | 29.0 | 31.1 |
| | 1000 | 512 | 3.8B | 33.9 | 30.2 | 32.1 |
| BranchNorm [33] | 200 | 512 | 863M | 34.2 | 28.5 | 31.4 |
| | 1000 | 512 | 3.8B | 35.0 | 29.6 | 32.3 |
| Transformer | 12 | 1024 | 466M | 34.0 | 27.6 | 30.8 |
| | 24 | 1024 | 618M | 34.9 | 28.1 | 31.5 |
| PCformer | 12 | 1024 | 466M | 36.0 | 29.1 | 32.6 |
| | 24 | 1024 | 618M | 36.9 | 30.5 | 33.7 |
| | 24 | 1536 | 1.2B | **37.7** | **32.2** | **35.0** |

**Sublayer Dropping** Additionally, we observe that our models benefit from the rich information brought by high-order predictor and subsequent implicit multistep corrector. To prevent from overfitting (settling into sub-optimal solutions) as the learning ability is quite strong, we borrowed the sublayer dropping technique [28, 34]. The drop rate is empirically set as 0.1 which delivers robust results in previous studies.

# 4  Experimental Results

We mainly evaluated the proposed method on machine translation, abstractive summarization, language modeling, and language understanding benchmarks. The details of datasets, and corresponding hyper-parameters please refer to Appendix C. For a clear comprehension, note that RK2-block (gated) is 2-order method with learnable coefficients in [29]'s work. And RK2-block (EMA) denotes our EMA strategy.

**Results of En-De and En-Fr** Table 1 compares the proposed PCformer with state-of-the-art systems in base and large configurations. As ODE Transformer is a strong baseline to ours, we implemented their results for a fair comparison. We can see that the proposed EMA coefficient learning method can further strengthen high-order methods, leading to better results than the gated fusion method in Li et al. [29]'s work (comparisons in RK2-block). And EMA can facilitate RK4-block to deliver a further gain of 0.40 BLEU points. The performance gains are more obvious for wider models, that PCformer sets or matches the new state-of-the-art with fewer parameters. Notably, a 6-layer PCformer (2-order) achieves a BLEU score of 30.90, surpassing the previous best of 30.77 by a 12-layer RK2-block with gated fusion [29]. For En-Fr, PCformer outperforms the standard Big model

Table 4: ROUGE results on CNN/DailyMail summarization dataset.

| Model | RG-1 | RG-2 | RG-L |
|---|---|---|---|
| Surface Connection [32] | 41.00 | 18.30 | 37.90 |
| Transformer [59] | 40.47 | 17.73 | 37.29 |
| RK2-block (gated) [29] | 41.58 | 18.57 | 38.41 |
| PCformer (2-order) | 41.96 | 18.99 | 38.74 |
| RK4-block [29] | 41.83 | 18.84 | 38.68 |
| PCformer (4-order) | **42.10** | **19.13** | **38.87** |

Table 5: Perplexity results on Wikitext-103. Adaptive refers to Adaptive Input Transformer [3].

| Model | Layers | Params | Valid | Test |
|---|---|---|---|---|
| Adaptive [3] | 8L | 146M | 21.11 | 21.00 |
| RK2-block (gated) [29] | 8L | 146M | 20.02 | 19.98 |
| PCformer (2-order) | 8L | 146M | 19.50 | 19.21 |
| Shortformer [48] | 8L | 146M | 19.04 | 19.78 |
| RK2-block (gated) [29] | 8L | 146M | 18.67 | 19.23 |
| PCformer (2-order) | 8L | 146M | 18.01 | 18.55 |

Table 6: PCformer results against Transformer++ [58] on various configurations. All models are trained on the same subset of the SlimPajama dataset (from 6B to 100B) with the Mistral tokenizer [21]. The last column shows the average over all benchmarks that use (normalized) accuracy as the metric.

| Model | Param | Tokens | Wiki. ppl ↓ | LMB. ppl ↓ | LMB. acc ↑ | PIQA acc ↑ | Hella. acc_norm ↑ | SciQ acc ↑ | ARC-c acc_norm ↑ | Wino. acc ↑ | Avg. acc ↑ |
|---|---|---|---|---|---|---|---|---|---|---|---|
| Transformer++ | 340M | 6B | 38.5 | 96.1 | 21.4 | 60.3 | 29.1 | 69.2 | 21.5 | 50.4 | 41.9 |
| PCformer | 340M | 6B | 35.3 | 78.8 | 23.6 | 61.6 | 30.1 | 71.6 | 22.9 | 51.8 | 43.6 |
| Transformer++ | 340M | 16B | 28.3 | 65.3 | 29.8 | 63.2 | 33.9 | 73.2 | 23.1 | 51.4 | 45.8 |
| PCformer | 340M | 16B | 25.6 | 39.7 | 34.5 | 65.2 | 36.9 | 79.6 | 23.2 | 52.2 | 48.6 |
| Transformer++ | 1.3B | 16B | 23.8 | 26.2 | 37.3 | 65.7 | 37.6 | 78.6 | 23.7 | 51.5 | 49.0 |
| PCformer | 1.3B | 16B | 20.9 | 23.2 | 42.5 | 68.3 | 43.4 | 81.5 | 25.1 | 52.4 | 52.2 |
| Transformer++ | 1.3B | 100B | 16.3 | 11.8 | 51.6 | 71.0 | 51.7 | 86.7 | 28.1 | 54.6 | 57.2 |
| PCformer | 1.3B | 50B | 16.2 | 9.4 | 55.1 | 71.9 | 54.8 | 88.6 | 29.6 | 57.2 | 59.5 |
| PCformer | 1.3B | 100B | 14.0 | 7.4 | 59.6 | 73.8 | 60.0 | 90.7 | 31.7 | 61.7 | 62.9 |
| PCformer | 3B | 50B | 13.6 | 6.5 | 62.1 | 74.4 | 61.9 | 90.6 | 32.4 | 61.9 | 63.9 |
| PCformer | 3B | 100B | **12.1** | **5.8** | **64.3** | **76.3** | **66.7** | **92.6** | **35.3** | **64.0** | **66.5** |

by 1.00 and 1.05 BLEU points with 2-order and 4-order configurations. This demonstrates that the predictor-corrector paradigm is a more parameter-efficient option than pure high-order methods.

**Results of En-Ro** Table 2 exhibits a similar phenomenon on the En-Ro task. Our predictor-corrector paradigm with EMA method achieves much better performance (35.49 *v.s.* 34.70) with `DeLight` within much less training cost. For a bigger model (line 7), it obtains a BLEU score of 35.80. A much higher performance (36.00) could be achieved by a carefully designed corrector which would be discussed in the subsequent analyses.

**Results of OPUS** Table 3 provides the comparison of PCformer against existing state-of-the-art models [69, 61] on the OPUS-100 testset. The findings here are three aspects: 1) Across all configurations, PCformer delivers significant BLEU gains over vanilla baselines. 2) Our 12-layer EMA Pre-Cor model attains an average SacreBLEU score of 32.6, which not only outperforms the 3.8B DeepNet but also beats its further optimized variant, BranchNorm, with only 1/8 model parameters. 3) PCformer can benefit from the enlarging width and depth. Notably, our 1.2B model shows an average SacreBLEU of 35.0, thereby setting a new state-of-the-art on the OPUS-100 testset.

**Abstractive Summarization** Table 4 presents the results of the abstractive summarization task. As shown, our PCformer consistently improves upon pure the high-order method [29], in terms of three rouge scores. Notably, PCformer (2-order) even beats RK4-block which consumes less computation cost. Additionally, PCformer (4-order) sets a new state-of-the-art on the summarization task which excludes models based on pre-trained models. This result strongly supports our hypothesis that an appropriate coefficient learning schedule is essential for the effectiveness of higher-order methods, thereby enhancing the performance of our PCformer model.

**Language Modeling** Table 5 presents a comparative analysis of our PCformer against vanilla Transformers in Adaptive Input Representation [3] and Shortformer [48] settings. Our 2nd-order configuration achieves significant reductions in perplexity (PPL), outperforming Adaptive and Shortformer by 1.79 and 1.23 PPL, respectively, even within identical model capacity constraints. Remarkably, PCformer surpasses the high-order method (RK2-block) by a substantial margin in both settings on both validation and test sets, demonstrating the superiority of PCformer.

Table 7: Comparison results on the GLUE development set.

| Model | CoLA Mcc | QQP Acc | MNLI-m/mm Acc | SST-2 Acc | STS-B Corr | QNLI Acc | RTE Acc | MRPC Acc | Avg. |
|---|---|---|---|---|---|---|---|---|---|
| BERT | 60.6 | 91.3 | 86.6/- | 93.2 | 90.0 | 92.3 | 70.4 | 88.0 | 84.0 |
| PCformer | 65.9 | 92.0 | 87.3/- | 93.6 | 90.8 | 92.8 | 74.7 | 91.5 | 86.1 |

**LM Evaluation Harness**   In response to the increasing significance of attention mechanisms in LLMs, we conducted a comprehensive evaluation of PCformer using established benchmarks, LM Evaluation Harness [13], focusing on a diverse range of downstream tasks including common-sense reasoning and question-answering. Table 6 presents our findings, where we utilized a llama-like model[3], Transformer++, as the foundation for PCformer. We trained models with parameter sizes ranging from 340M to 3B, using datasets comprising 6B to 100B tokens from Slimpajama. The experimental results indicate that PCformer consistently surpasses the performance of a well-tuned Transformer of equivalent capacity. Notably, PCformer achieves an average score improvement of 1.7 points for the 340M model and 5.7 points for the 1B model across six challenging subtasks. When scaled to a 3B parameter size, PCformer demonstrates even greater gains, achieving an additional 3.5 average score improvement compared to the 1B model, underscoring its scalability and potential with larger model capacities and richer training datasets.

**Language Understanding**   We also validate our method on the widely used natural language understanding benchmarks, namely GLUE, which consists of 8 sub downstream tasks. The evaluation metrics are as follows: The result for STS-B is the Pearson correlation; Matthew's correlation is used for CoLA; Other tasks are measured by Accuracy. The results are presented in Table 7. We can see that PCformer achieves 2.1 points (on average) improvement over the BERT-large, which demonstrates the effectiveness of PCformer.

# 5   Analysis

In this section, we explore several significant issues, comprising the visualization of truncation errors, COMET results, and a set of essential ablation studies. We primarily conducted ablation studies on machine translation tasks, but the conclusions are generalizable.

**Quantization of the Truncation Error**   Following the suggestion in [29]'s work, we use the perplexity between the single-layer Transformer decoder output and the ground truth to approximate the "truncation error". The results of Table 8 were conducted on the Penn Treebank dataset. We see that the proposed EMA method achieves a lower perplexity than the learnable coefficient (gated) learning method, similar observation in 4-order (EMA *v.s.* Rk4-block). Additionally, the Predictor-Corrector paradigm can further reduce the truncation error, which demonstrates the effectiveness of our method.

Table 8: Comparison of PPL on PTB.

| Model | 1-Layer | 2-Layer |
|---|---|---|
| Residual-Block | 142.33 | 136.07 |
| RK2-block | 131.80 | 123.12 |
| RK2-block (gated) [29] | 128.48 | 121.02 |
| RK2-block (EMA) | 124.01 | 119.65 |
| PCformer (2-order) | 120.91 | 118.37 |
| RK4-block | 126.89 | 119.46 |
| RK4-block (EMA) | 121.82 | 116.77 |
| PCformer (4-order) | **119.27** | **114.32** |

**Evaluation by COMET**   Our PCformer consistently outperforms baselines, showing an even larger gap in COMET than BLEU, as shown in Table 9. Both metrics exhibit similar performance trends, highlighting our approach's effectiveness. Increasing model depth from 6 to 12 layers does not improve BLEU for the En-De task but results in a 0.64 COMET gain. A similar pattern is observed in the En-Fr task, where PCformer (4-order) achieves comparable BLEU to its 2-order counterpart but gains 0.24 in COMET.

Table 9: COMET (%) *v.s.* BLEU (%) results.

| Model | En-De | | En-Fr | |
|---|---|---|---|---|
| | BLEU | COMET | BLEU | COMET |
| Transformer-big (6L) | 29.21 | 51.87 | 42.89 | 71.21 |
| PCformer (RK2) | 30.90 | 54.74 | 43.85 | 73.96 |
| PCformer (RK4) | - | - | 44.10 | 74.76 |
| Transformer-big (12L) | 29.91 | 52.90 | 43.22 | 72.33 |
| PCformer (RK2) | 30.95 | 55.38 | 44.27 | 75.09 |
| PCformer (RK4) | - | - | 44.21 | 75.33 |

---

[3] we followed the setting in Gu and Dao [14]'s work.

Table 10: Ablation on the several choices of the predictor and corrector on four translation tasks.

| Predictor | Corrector | En-De | En-Fr | En-Ro | OPUS |
|---|---|---|---|---|---|
| First-order Baseline | - | 29.91 | 43.22 | 34.20 | 31.5 |
| ODE Transformer | - | 30.77 | 43.96 | 35.28 | 32.3 |
| RK2-block with EMA | Multistep Method | 30.70 | **44.27** | 35.55 | **33.7** |
| RK2-block with EMA | Backward Euler Method | **30.95** | 43.68 | **36.00** | 33.2 |
| Multistep Method | Multi-step Method | 30.30 | 43.92 | 35.30 | 33.0 |
| Multistep Method | RK2-block with EMA | 29.78 | 42.68 | 34.40 | 32.5 |
| Multistep Method | Backward Euler Method | 30.30 | 43.62 | 35.27 | 32.8 |

Table 11: Ablations on the PCformer design on the machine translation task. The evaluation metric is BLEU (%).

| Model | $\gamma$ | BLEU |
|---|---|---|
| Transformer-base | - | 27.89 |
| RK4-block (EMA) | 0.25 | 28.90 |
| RK4-block (EMA) | 0.50 | **29.43** |
| RK4-block (EMA) | 0.75 | 28.99 |
| RK4-block (EMA) | 0.99 | 29.20 |

(a) **Coefficients**: the effect of different $\gamma$ for the EMA coefficient learning.

| Model | BLEU |
|---|---|
| Transformer-big | 29.21 |
| RK2-block (EMA + Pre-Cor) | 30.95 |
| w/o RK-Norm | failed |
| Layer-wise $\gamma$ | 30.63 |
| Vector $\gamma$ | 30.88 |

(b) **Other techniques**: Figuring out several key components for high-order solutions.

**Ablation Study on Predictor-Corrector Framework**    The predictor-corrector framework is crucial in our work, with the choice of ODE solutions for each component significantly impacting performance. We experimented with various combinations of predictors and correctors, including high-order methods, linear multistep methods, and the Backward Euler method. Table 10 summarizes these results. We chose RK2-block with EMA as the default for high-order solutions due to its performance and efficiency. Key insights include: 1) The predictor must be highly accurate, as it sets the performance lower bound. High-order predictors outperform the multistep method (DLCL) and the Euler method. 2) A complex corrector isn't always optimal; a Backward Euler method suffices for small and medium datasets (e.g., En-Ro and En-De), while more complex methods may cause overfitting. 3) Combining a multi-step method predictor with a high-order corrector performed worse than other combinations, highlighting the importance of predictor choice.

**Ablation Study on Core Design Technique**    Table 11 presents the performance of our PCformer with various initial coefficients for the EMA method and stable training techniques. Our default, with $\gamma$ set to 0.5, performs best because smaller $\gamma$ values disrupt the numerical bound, and larger values overly focus on recent approximations. As detailed in Section 3.2, RK-Norm is essential for training stability, as shown by the BLEU score drop without it. Testing different layer-wise coefficients and replacing the learnable scalar $\gamma$ with a learnable matrix vector showed no significant performance difference, so these were excluded from our default settings.

**Inference Speed and Memory Consumption**    Table 12 presents a detailed comparison of inference speed and memory consumption across various large model configurations, revealing that the proposed PCformer models achieve satisfactory inference performance. This is primarily because the computational overhead is concentrated on the decoder side rather than the encoder, as demonstrated in our experiments. Additionally,

Table 12: Comparison of inference speed (sentences/s) and memory consumption (GB) between the vanilla Transformer and numerical Transformers.

| Model | Layers | Inference | Memory | BLEU |
|---|---|---|---|---|
| Transformer | 6 | 98.7 | 13.2 | 29.2 |
| Transformer | 12 | 94.5 | 18.7 | 29.7 |
| Transformer | 24 | 87.3 | 23.5 | 29.8 |
| ODE Transformer (RK2) | 6 | 93.5 | 15.1 | 30.7 |
| PCformer (RK2 predcitor) | 6 | 90.3 | 16.2 | 30.9 |
| ODE Transformer (RK4) | 6 | 87.1 | 17.3 | 30.5 |

PCformer is memory-efficient, as shown by the memory usage comparison between the baseline and the ODE Transformer. Despite the fact that our PCformer models are more than twice as slow as the vanilla baseline in encoder-only and decoder-only configurations, they deliver significantly superior performance, making the trade-off worthwhile. These performance gains are clearly illustrated in Table 6, where the substantial improvement in model effectiveness justifies the increased inference time. We acknowledge that further optimization to accelerate PCformer's inference speed is a promising direction for future research.

**More Analyses**   Due to the limited space in the main content, we summarized more detailed analyses in Appendix D, including the parameter efficiency (Figure 3), illustration of training and validation curves (Figure 4), and visualization of coefficients during the learning procedure (Figure 5). We anticipate that these analyses will offer a deeper and more comprehensive understanding of our method.

# 6   Related Work

**Ordinary Differential Equations**   The connection between ResNet and ODEs was first proposed by Weinan [63], while Neural ODENet [8] introduced a new perspective on neural architecture design. Several architectures [71, 24, 37, 17, 72, 38, 53] can be interpreted from the ODE perspective. Recent studies leverage ODE benefits for Transformers. Lu et al. [38] proposed MacaronNet using the Strang-Marchuk Splitting Scheme, and Zhang et al. [70] introduced continuous self-attention models. Dutta et al. [12] redesigned Transformer architecture for efficiency from a multi-particle dynamic system view. Li et al. [29] showed that first-order ODE blocks could cause error accumulation, and high-order methods were suggested as solutions. In this work, we advance the Transformer design with a more accurate Predictor-Corrector paradigm and a general coefficient learning strategy inspired by the exponential moving average, showing significant performance improvements on NLP benchmarks.

**ODE and Diffusion Models**   ODEs and numerical methods are also popular in diffusion models, reducing prediction errors in denoising processes. Text-to-image generation typically uses a two-stage model, including a text-to-image diffusion model and super-resolution models. The standard diffusion model, DDPM [19], requires up to 1000 iterations to recover images from Gaussian noise. Subsequent work accelerated DDPMs using denoising equations [57] or scheduled variance [44], though often at the cost of performance. Liu et al. [31] proposed treating DDPMs as solving differential equations on manifolds, introducing a pseudo linear multi-step method for efficiency and performance. Further diffusion acceleration efforts [35, 36] were motivated by ODE benefits. More recently, Xu et al. [67] presented a deep generative model solving the Poisson equation, opening new possibilities in text-to-image generation. ODEs also show promise in discrete diffusion models, as demonstrated by Lezama et al. [26], who used a predictor-corrector paradigm to enhance the accuracy.

# 7   Conclusions

This paper advances the design of parameter-efficient neural network backbones through a numerical analysis perspective. Previous work has utilized high-order ODE solutions for more accurate approximations at each block, yielding promising results on various sequence generation tasks. However, challenges remain, such as the scalability of learnable coefficients to RK4-blocks and the lack of exploration into implicit ODE methods. To address these issues, we introduce a predictor-corrector framework to improve estimation precision. Additionally, we proposed an EMA coefficients learning strategy to promote coefficients learning for high-order methods with high flexibility. Experimental results across 8 benchmarks demonstrate the general ability and strong effectiveness of our approach. More concretely, Our PCformer achieves 30.95 and 44.27 BLEU scores on the WMT'14 En-De and En-Fr, setting a new state-of-the-art result on both testsets without considering data augmentation methods. Also, it delivers an average BLEU of 35.0 on the OPUS multilingual dataset, beating Deep-Net and other variants with much fewer model parameters. Notably, PCformer demonstrates strong potential in large language model scenarios, outperforming vanilla LLama models by a considerable margin across various configurations on LM Harness Evaluation. Our codebase could be found at `https://github.com/libeineu/PCformer`.

# Acknowledgments

This work was supported in part by the National Science Foundation of China (No.62276056), the Natural Science Foundation of Liaoning Province of China (2022-KF-26-01), the Fundamental Research Funds for the Central Universities (Nos. N2216016 and N2316002), the Yunnan Fundamental Research Projects (No. 202401BC070021), and the Program of Introducing Talents of Discipline to Universities, Plan 111 (No.B16009). Jingang Wang is funded by Beijing Nova Program (Grant NO. 20220484098).

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

# Supplementary Material for "Predictor-Corrector Enhanced Transformers with Exponential Moving Average Coefficient Learning"

## A  Broader Impact

We do not anticipate any specific negative impacts from our work. However, as with any machine learning method, we recommend exercising caution. Our primary contribution is advancing neural model design from a numerical perspective, aiming to facilitate models' parameter learning and enhance the performance, which we believe to be environmentally friendly. This approach encourages the community to enhance neural models by integrating knowledge from various domains, such as physiology, physics, and mathematics, thereby further benefiting AI research.

## B  Limitations and Future Work

The proposed PCformer has demonstrated significant performance gains across a variety of tasks, including sequence generation, natural language understanding, and language modeling. While the encoder-decoder PCformer is computationally efficient, as it only employs the predictor-corrector paradigm on the encoder side, the primary computational overhead during inference remains in the decoder. However, when our method is applied to encoder-only models (for NLU) or decoder-only models (for LM/LLM), the additional computational complexity becomes non-negligible. Addressing how to further accelerate inference in these scenarios remains critical, and we aim to explore this in future work.

We plan to address this issue from two primary aspects:

- High-order Computation in Latent Space: When adopting high-order methods, the iterative computations among inner steps cannot be skipped. Therefore, we aim to investigate whether high-order computations can be performed in a latent space with a reduced dimensionality, such as 64 or 128 hidden dimensions, compared to the original 1024 dimensions. This approach presents a significant theoretical challenge: maintaining the stability of the ODE while computing higher-order intermediate approximations in the latent space.

- High-order Training and Inference in a First-order Manner: Another alternative is to achieve high-order training and inference using a first-order (Euler-like) approach. This might involve employing distillation techniques or treating high-order computations as a form of training regularization.

We will continue to advance the design of PCformers to balance performance gains with inference efficiency, particularly in the context of large language models. This research could provide valuable insights to the community on whether such modeling methods can further enhance the performance of LLMs.

## C  Dataset and Evaluation

### C.1  Machine Translation

**Datasets**  We present experimental results across three WMT benchmarks, including WMT'14 English-German (En-De), WMT'14 English-French (En-Fr), and WMT'16 English-Romanian (En-Ro), as well as on a large-scale, challenging multilingual machine translation benchmark (OPUS).

- For the En-De task, the training data consisted of approximately 4.5M tokenized sentence pairs, as in [59]. All sentences were segmented into sequences of sub-word units [54] with 32K merge operations using a shared vocabulary. We selected *newstest2013* as the validation data and *newstest2014* as the test data.

- For the En-Fr task, we used the dataset provided within Fairseq, i.e., 36M training sentence pairs from WMT'14. *newstest2012+newstest2013* was the validation data and *newstest2014* was the test data.

- For the En-Ro task, we replicated the setup of [39], which used 600K/2K/2K sentence pairs for training, evaluation and inference, respectively.

- Apart from the aforementioned three bilingual translation benchmarks, OPUS is a multilingual benchmark that contains more challenges for the model to serve. We choose OPUS [69], an English-centric corpus covering 100 languages, which is randomly sampled from the OPUS collection. For a fair comparison with prior work,

Table 13: Statistics of the datasets and hyperparameters for sequence generation tasks. For the dataset, we both report the vocabulary size, sentence numbers of training, validation and test sets. For the training, Lr denotes the peaking learning rate and Warmup denotes the warmup step of the Adam optimizer. WD denotes whether we applied word dropout. For the inference, Beam and LP denote the beam size and length penalty, respectively.

| Benchmarks | Vocab | Dataset | | | Training | | | | Inference | |
|---|---|---|---|---|---|---|---|---|---|---|
| | | Train | Dev | Test | Lr | Warmup | Batch | Steps | Beam | LP |
| WMT'14 En-De | 34040 | 4.5M | 3000 | 3003 | 0.002 | 16000 | 80K | 50K | 4 | 0.6 |
| WMT'14 En-Fr | 37284 | 35.7M | 26,822 | 3003 | 0.002 | 16000 | 320K | 100K | 4 | 0.6 |
| WMT'16 En-Ro | 34976 | 602K | 1999 | 1999 | 0.002 | 8000 | 80K | 17K | 5 | 1.3 |
| OPUS | 209816 | 109.3M | 371068 | 376000 | 0.002 | 16000 | 320K | 100K | 5 | 1.0 |
| CNN/DailyMail | 32584 | 287K | 13368 | 11490 | 0.002 | 8000 | 160K | 50K | 4 | 2.0 |
| PTB | 10008 | 908 | 73 | - | 0.002 | 2000 | 160K | 3K | 4 | 2.0 |
| Wikitext-103 | 32584 | 180K | 3760 | 11490 | 0.002 | 8000 | 160K | 50K | 4 | 2.0 |

we used the already sampled subset[4] and following the script provided by Zhang *et al.* [69] to pre-process the data, including the data filtering, sentence piece training and applying. After that, OPUS-100 contains approximately 55M sentence pairs. Following the same training strategy with [69, 61], we train a single model for both to English (XE) and from English (EX), thus the total training data is 110M. Note that 2000 sentence pairs for each language to serve as validation and test sets.

**Evalutation** We measured performance in terms of BLEU. Both tokenized BLEU and SacreBLEU[5] scores were reported on the En-De and En-Fr tasks. For the OPUS task, we average the SacreBLEU scores among 94 languages for both X→En and En→X. The beam size and length penalty of each task are summarized in Table 13. In order to attain results that are more compelling, it is imperative to acknowledge that BLEU may not be a suitable metric for evaluating a model's performance. To supplement our evaluation, we have included a summary of the COMET scores [49] of the top-performing model, utilizing a reference-based model[6].

**Training Details** In accordance with the recommendations provided by [27], we have incorporated relative positional representation [55], namely RPR for short, into our model architecture to establish stronger baselines. To ensure stable learning during training, we have also borrowed the merit of dense connections among layers [62] for stable optimization within FP16 training. Our models were trained on 8 GPUs with 4,096 tokens per GPU. For the En-De, En-Fr and OPUS tasks, we have observed that larger batching schemas often result in better convergence [45]. Therefore, we accumulated gradients every 2 and 8 steps for En-De and En-Fr/OPUS, respectively. Adam optimizer [22] with $(0.9, 0.997)$ for $\beta_1$ and $\beta_2$ is adopted. The hyperparameters including the learning rate, the warmup step and the total training steps of three tasks could be found in Table 13. Note that we trained Base/Deep and Big models for 50K and 100K steps on the En-De task. We regarded merging SAN and FFN as the default ODE block. In addition, main results were the average of three times running with different random seeds (1, 42 and 2024), and we averaged the last several checkpoints towards the robustness. The detail of Base/Deep/Wide configurations is as follows:

- Base/Deep Model. The hidden size of self-attention was 512, and the dimension of the inner-layer in FFN was 2,048. We used 8 heads for attention. For training, we set all dropout to 0.1 as default, including residual dropout, attention dropout, ReLU dropout. Label smoothing $\epsilon_{ls} = 0.1$ was applied to enhance the generation ability of the model. For deep models, we only enlarged the encoder depth considering the inference speed.

- Wide (or Big) Model. We used the same architecture as Transformer-Base but with a larger hidden layer size 1,024, more attention heads (16), and a larger feed forward inner-layer (4,096 dimensions). The residual dropout was set to 0.3 for the En-De task and 0.1 for the En-Fr task.

## C.2 Abstractive Summarization

We also tested the models' ability to process long sequences on the CNN-DailyMail summarization task [42, 18]. The preprocessed method was the same as in [46]. We used a shared BPE with 30K operations, resulting in a vocabulary of 32,580 entries. The evaluation metric was F1-Rouge [30] (Rouge-1, Rouge-2 and Rouge-L).

---

[4]https://object.pouta.csc.fi/OPUS-100/v1.0/opus-100-corpus-v1.0.tar.gz.

[5]BLEU+case.mixed+numrefs.1+smooth.exp+tok.13a+version.2.3.1

[6]The default setting which uses a reference-based regression model built on top of XLM-R (Large).

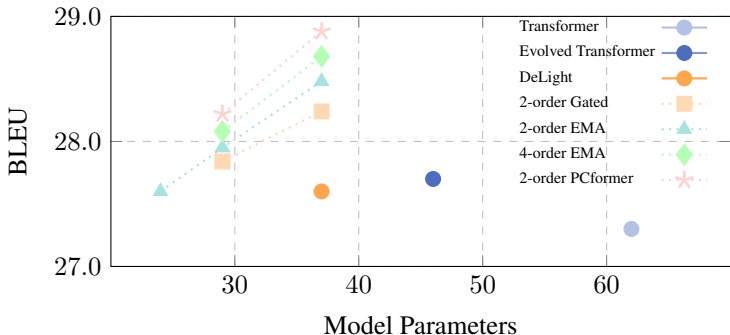

Figure 3: The comparison of BLEU as well as model capacities and training costs against previous state-of-the-art deep transformers.

## C.3 Language Modeling

**Datasets**  As mentioned above, the truncation error analysis is conducted on the Penn Treebank [41], which is a widely-used language model dataset. It contains 88K, $3,370$ and $3,761$ sentences for training, validation and test. The vocabulary size is 10K. We set the layer depth of the language model to 1 or 2 to make a fair comparison. Assuming the layer depth is 1, then the loss between the model output and the true label could be regarded as the truncation error. In this way, we alleviate the influence of the error accumulation across different layers. Apart from PTB, we also evaluate our approach on another widely acknowledged language modeling dataset, Wikitext-103 dataset [40], which is the largest available word-level language modeling benchmark with long-term dependency. WikiText-103 consists of 103M training tokens from 28K articles on Wikipedia, and the average length of tokens per article is about 3.6K. The data is can be easily obtained and preprocessed following Baevski *et al.* [3]'s work.

**Setups**  For the PTB dataset, we used the `transformer_base` configuration, whose hidden size is 512, and the filter size of the FFN is $2,048$. All the dropout rates are 0.1, including the residual dropout, attention dropout and ReLU dropout. Each model was trained up to 20 epochs, while most models arrived at convergence on the validation set when the epoch is 10. Then the validation PPL began to increase, though the training PPL is still declining. This is due to the small model capacity. The warmup step was $2,000$ and the batch size was $4,096$. The max learning rate was set to 0.0007. For the Wikitext-103 dataset, all models were based on the original open-source Fairseq toolkit, and the corresponding configuration is `transformer_lm_baevski_wiki103`. It inherits the configuration of `transformer_big` with $1,024$ hidden size and $4,096$ filter size. Besides these common hyper-parameters, it adopts the adaptive input representation to reduce the embedding matrix by decreasing the embedding size of low-frequency words or sub-words. This is also the most common choice when building large-scale language models.

## C.4 Language Understanding

Our proposed PCformer by continue training the BERT model using the provided pretrained model, *BERT-Large-cased*. And we evaluated on the General Language Understanding Evaluation (GLUE) [60]. Concretely, the GLUE benchmark has 8 different text classification or regression tasks including MNLI, MRPC, QNLI, QQP, RTE, SST-2, SST-B, and CoLA.

## C.5 LM Evaluation Harness

We also evaluate our PCformer on a wide range of downstream tasks covering common-sense reasoning and question-answering, including LAMBADA (LMB. [47]), PIQA [4], SciQ [64], Winograde (Wino. [52]), HellaSwag (Hella. [68]), and ARC-challenge [9]. We mainly evaluate language models via zero-shot evaluation. We randomly sampled 100B tokens from Slimpajama dataset for training.

## D  More Analyses

**Parameter Efficiency**  Figure 3 summaries the results of several efficient Transformer variants, including Lite Transformer [66], DeLight [39], a light version of the Evolved Transformer [56], and our PCformer. It is clear to see that PCformer is significantly more parameter efficient than others. We make detailed comparisons of PCformer within different hyper-parameters, comprising of hidden size and model depth. Concretely, RK2-block

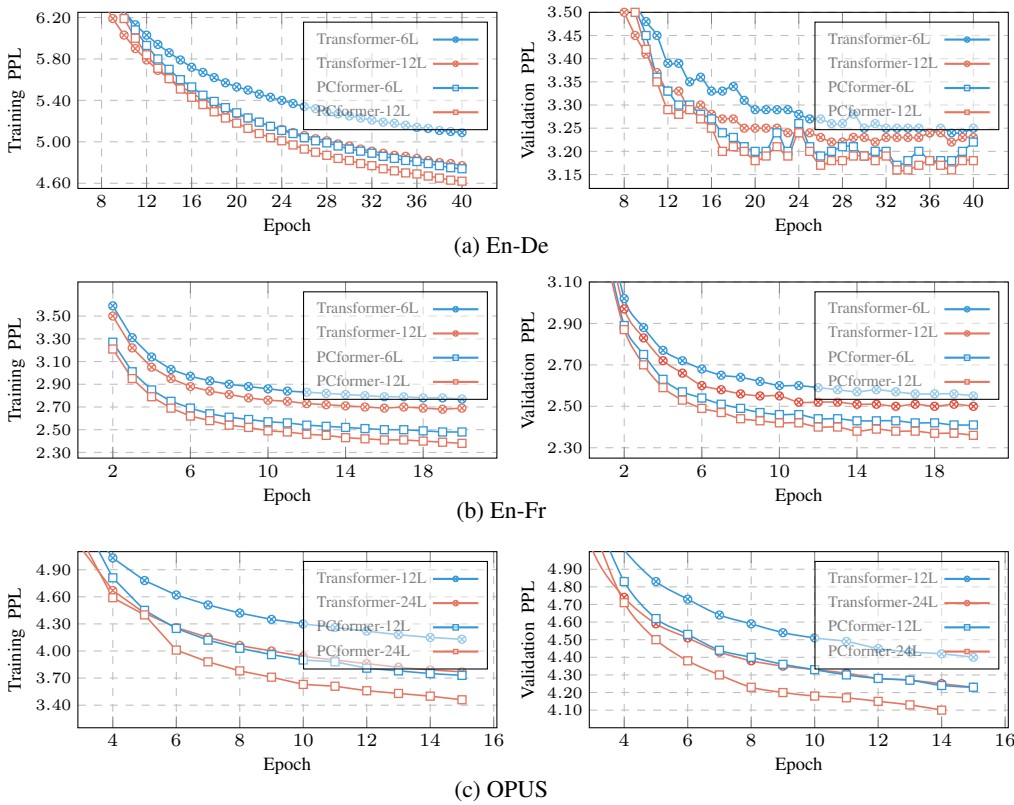

Figure 4: The comparison of training and validation PPL on base and wide models.

with gated fusion coefficient learning strategy delivers stronger performance than `DeLight` within the same model parameters. And it is on par with `DeLight` in terms of BLEU, but having 9M fewer parameters. Moreover, as expected, our newly proposed EMA coefficient learning method slightly improves the performance of gated fusion in almost all scenarios. A notable bonus it brought is higher-order solutions, e.g., RK4-block is superior to RK2-block with the help of EMA. It may offer a new choice for deploying NMT systems on edge devices.

**Training and Validation Perplexity** Apart from the BLEU scores, we also compare our methods with baselines regarding perplexity on both training and validation sets. Figure 4 plots the training and validation PPL curves of the PCformer (RK2-block (EMA + Pre-Cor)) and the baseline on two representative translation tasks. All models were in big configurations for more convincing conclusions. The proposed RK2-block with EMA coefficient learning and Predictor-Corrector framework delivers much lower training and validation PPLs within the same configurations. More specifically, our method still benefits from increasing model depth, as the 12-layer model outperforms the 6-layer one. For both the En-De and En-Fr tasks, we observed that our 6-layer method even shows lower PPLs than a 12-layer Transformer. This phenomenon is more evident in the WMT En-Fr task, which again demonstrates the high parameter efficiency of our PCformer.

**Visualization of the Coefficient Learning Procedure** We also collect the learning process of learnable coefficients $\gamma$ during training. As we discussed above that the inspiration of EMA coefficient learning method comes from the prior that the most current approximation is more precise. This assumption has already been clarified by the comparisons of truncation errors. Here, we want to figure out how coefficients learned if removing the constraint. To achieve this goal, we set all coefficients to be independently initialized by the mean of 1, e.g., a 2-order block with $\gamma_1 = 0.5, \gamma_2 = 0.5$. Figure 5 plots the learning curves of RK2-block and RK4-block within these two learning strategies. As we can see that, for the independent initialization scenarios, coefficients vary dramatically within the first several epochs, and then show convergence in a small range. Both two figures show that the contribution of the most current coefficient is larger than others.

To our surprise, $\gamma_2$ in RK2-block (Independent) converges to large than 1 and oppositely, $\gamma_1$ is even goes to a negative value. Meanwhile, $\gamma_3$ in the RK4-block (Independent) shows a negative impact to the final solution, but the other 3 coefficients vary within our expectation. We notice the order relation among these 4 coefficients are consistent with the numerical suggested coefficients in the Adams-Bashforlth method (-9/24, 37/24, -59/24,

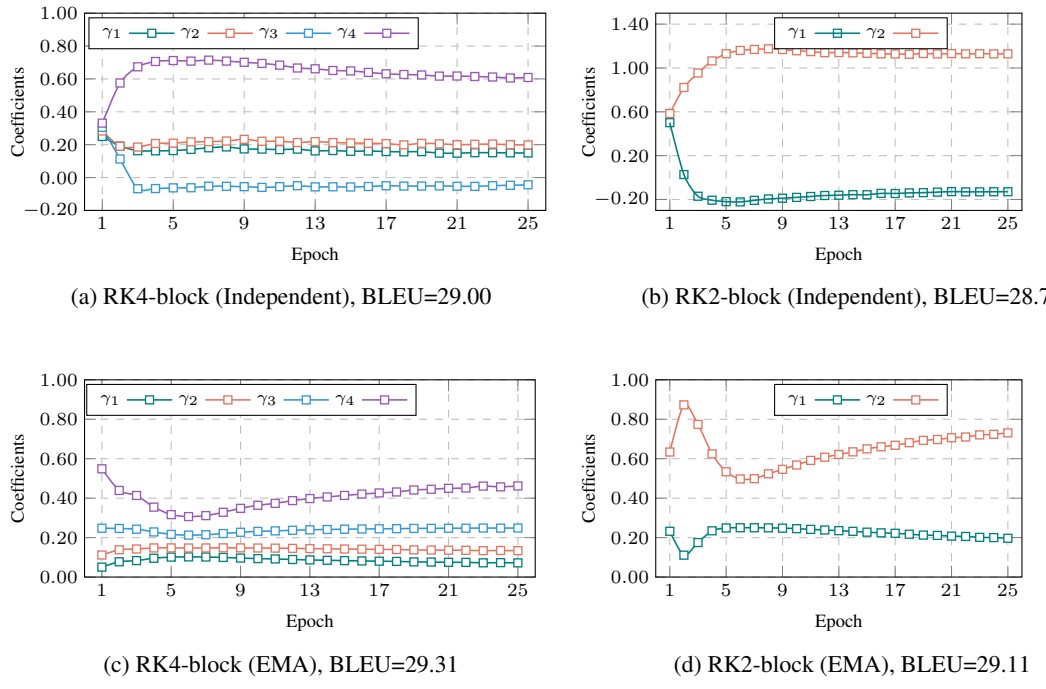

(a) RK4-block (Independent), BLEU=29.00

(b) RK2-block (Independent), BLEU=28.78

(c) RK4-block (EMA), BLEU=29.31

(d) RK2-block (EMA), BLEU=29.11

Figure 5: The coefficient learning curves of independent initialization and EMA oin both 2-order and 4-order scenarios. The experiments are conducted on WMT En-De.

55/24). This indicates the underlying relationship between the numerical analysis and the neural network optimization. While, after employing our EMA method, coefficients are optimized along our expected direction, and these models are empirically better than those without constraints.

**Results on Time-Series Forecasting**    We also followed the reviewer's suggestion to evaluate PCformer on time-series forecasting tasks. We selected 10 multivariate datasets from UEA Time Series Classification Archive following the setting and the codebase provided by Flowformer [65]. Thus we choose the Flowformer as the baseline, which is also a strong model on these testsets. For the details, we build the PCformer upon Flowformer and report the 2-order predictor and Euler corrector as the training data is very small. Also, we use RK-Norm to avoid the model suffering from the overfitting problem as the authors of Flowformer trained their models upon to 100 epochs (or even 400 epoch on some tasks.). The results are evaluated by the best accuracy. We can see that PCformer can beat the Flowformer by 2 average score on 10 testsets, which demonstrates the effectiveness on time-series forecasting tasks.

Table 14: Comparison of Flowformer and PCformer on different datasets.

| Dataset | Flowformer | PCformer |
|---|---|---|
| ETHANOLCONCENTRATION | 30.3 | 33.9 |
| FACEDETECTION | 67.0 | 68.2 |
| HANDWRITING | 29.1 | 33.5 |
| HEARTBEAT | 77.0 | 78.5 |
| JAPANESEVOWELS | 98.4 | 99.2 |
| PEMS-SF | 87.2 | 87.9 |
| SELFREGULATIONSCP1 | 89.0 | 92.2 |
| SELFREGULATIONSCP2 | 55.0 | 56.1 |
| SPOKENARABICDIGITS | 98.0 | 100.0 |
| UWAVEGESTURELIBRARY | 85.3 | 86.3 |
| **Average Score** | 71.6 | 73.6 |

