# OpenReview forum: "Predictor-Corrector Enhanced Transformers with Exponential Moving Average Coefficient Learning"
_NeurIPS.cc/2024/Conference — NeurIPS 2024 poster_

### Official Review · Reviewer_hvMK · 2024-07-11

**Soundness:** 3
**Presentation:** 3
**Contribution:** 3
**Rating:** 7
**Confidence:** 3

**Summary:**

This work presents a transformer architecture inspired by predictor-corrector methods for solving ODE problems. It adopts the Adams-Bashforth-Moulton method and utilizes an exponential moving average (EMA) method to compose the predictor, discussing two correctors (the EMA-based and the simple backward Euler method). Without the background of ODE, the proposed method could be seen as a special case of a transformer with cross-layer connections. Consider that the residual connection only links two layers, while the proposed method links multiple layers in a high-order way.

**Strengths:**

This work provides theoretical results to enhance the connections between residual connections and the ODE solver, providing an interesting perspective on viewing cross-layer connections within neural networks.

Using EMA as a high-order predictor is a simple and flexible solution, and the experiments demonstrate its effectiveness.

**Weaknesses:**

The experiments are insufficient and out-of-date. As the main contribution is the new transformer architecture, scaling law style experiments and testing on modern LLMs' benchmarks are recommended. I saw that Appendix D has some results, but it's better to have a formal experiment.

**Questions:**

What's the efficiency of the proposed method, such as inference time and GPU memory consumption? I saw that inference efficiency has been mentioned in the limitations; it's better to have quantitative results and also memory states.

**Limitations:**

Agree with the limitation section.

---

> ### Author Rebuttal · Authors · 2024-08-06
>
> Following your suggestion, we have scaled up our large language model (LLM) experiments, focusing on how our PCformer performs in the LLM setting. We aim to demonstrate the capabilities of PCformer from both model capacity and data volume perspectives. The updated results are as follows:
>
> | Model(Params & Tokens)     | Wiki.(ppl) | LMB.(ppl) | LMB.     | PIQA     | Hella.   | SCIQ     | ARC-c    | Winograde| Avg.     |
> | -------------------------- | ---------- | --------- | -------- | -------- | -------- | -------- | -------- | -------- | -------- |
> | Transformer++ (340M & 6B)  | 38.5       | 96.1      | 21.4     | 60.3     | 29.1     | 69.2     | 21.5     | 50.4     | 41.9     |
> | PCformer (340M & 6B)       | 35.3       | 78.8      | 23.6     | 61.6     | 30.1     | 71.6     | 22.9     | 51.8     | 43.6     |
> | Transformer++ (340M & 16B) | 28.3       | 65.3      | 29.8     | 63.2     | 33.9     | 73.2     | 23.1     | 51.4     | 45.8     |
> | PCformer (340M & 16B)      | 25.6       | 39.7      | 34.5     | 65.2     | 36.9     | 79.6     | 23.2     | 52.2     | 48.6     |
> | Transformer++ (1.3B & 16B) | 23.8       | 26.2      | 37.3     | 65.7     | 37.6     | 78.6     | 23.7     | 51.5     | 49.0     |
> | PCformer (1.3B & 16B)      | 20.9       | 23.2      | 42.5     | 68.3     | 43.4     | 81.5     | 25.1     | 52.4     | 52.2     |
> | Transformer++(1.3B & 100B) | 16.3       | 11.8      | 51.6     | 71.0     | 51.7     | 86.7     | 28.1     | 54.6     | 57.2     |
> | PCformer (1.3B & 50B)      | 16.2       | 9.38      | 55.1     | 71.9     | 54.8     | 88.6     | 29.6     | 57.2     | 59.5     |
> | PCformer (1.3B & 100B)     | **14.0**   | **7.46**  | **59.6** | **73.8** | **60.0** | **90.7** | **31.7** | **61.7** | **62.9** |
>
>
> - In our initial submission, we reported results on a 340M parameter configuration trained with 6B Slimpajama data. Here we included additional perplexity metrics on Wikitext and Lambada to provide more comprehensive results. Furthermore, for more convincing results, we scaled the model size from 340M to 1.3B parameters, which is the maximum size feasible within the limited time. Scaling to larger models requires additional engineering efforts, such as implementing tensor or pipeline parallelism for robust and efficient training (using Megatron codebase). We plan to report results on 7B or larger models in our next version and we believe PCformer can indeed perform strongly on those settings.
>
> - For the data, we scaled from 6B to 16B and 100B tokens. Our findings show that with more training data, the average performance improves significantly. This demonstrates that our model benefits from increased data size, without experiencing diminishing returns. This indicates that model bias persists and plays a crucial role in our setting (Larger settings still worth to be explored in future work). As the model size increases (from 340M to 1.3B), PCformer shows substantial performance gains, both in terms of accuracy on the LM harness evaluation and lower perplexity on Wikitext and Lambada tests.
>
> - We trained the 1.3B model on a cluster of 256 A100 GPUs. Note that 1.3B models consist of 24 layers, where the hidden size is 2048 and the FFN size is 5432 （8/3 * hidden size, 16 attention heads and SiLU activation functions. The baseline (1.3B + 100B tokens) was trained up to 20 hours and nearly 40 hours for our PCformer. Thus PCformer (1.3B + 50B tokens) was trained within similar time.
>
> - Additionally, given that PCformer consumes more FLOPS per forward pass, we compared models with similar total FLOPS consumption. Our PCformer trained on less than 50B data outperformed Transformer++ trained on 100B data by a significant margin. Moreover, PCformer (340M & 16B) achieved results comparable to Transformer (1.3B & 16B) with nearly 1/4 of the parameters. These results demonstrate that PCformer remains competitive in settings with larger model capacity and data volumes, highlighting the potential for further research in model designs that fully utilize available data.
>
> - These results demonstrate that PCformer remains competitive in settings with larger model capacity and data volumes, highlighting the potential for further research in model designs that fully utilize available data. While the increased inference and training costs in the decoder-only paradigm are notable, the substantial performance gains justify continued exploration of PCformer, including parameter-efficient training algorithms.
>
> - We believe these updated results address concerns about the relevance and sufficiency of our experiments. Thank you for your valuable feedback. Please let us know if you have any further concerns.
>
>
>
> > Q1: What's the efficiency of the proposed method, such as inference time and GPU memory consumption? I saw that inference efficiency has been mentioned in the limitations; it's better to have quantitative results and also memory states.
>
> This question is a common issue among reviewers, we hope you can see the global response W1 for more details!
>
> Due to the limited page in global response, we provide a comparison of the total inference time between PCformer and the baseline in the 1.3B setting (trained on 100B tokens). Using 8 A100 GPUs with CUDA 12.2, we measured the inference time across all benchmarks listed in the table. The PCformer took 1428 seconds, while the Transformer++ took 1037 seconds. Although PCformer's inference time is longer, the performance gains make the additional time investment worthwhile.

---

### Official Review · Reviewer_MB44 · 2024-07-12

**Soundness:** 3
**Presentation:** 3
**Contribution:** 3
**Rating:** 6
**Confidence:** 4

**Summary:**

This paper takes inspiration from established high-order approaches in numerical analysis for solving differential equations to improve the architectural design of Transformers. Prior work has shown that residual networks can be seen as discrete approximations of Ordinary Differential Equations (ODE) and explored methods to improve the quality of the solution. Specifically, this paper introduces a predictor-corrector learning framework to minimize approximation errors and an exponential moving average-based coefficient learning method. These advancements are used within Transformer architectures that are evaluated on several tasks such as translation, summarization, language modeling and language understand tasks, improving over standard and previous ODE-based Transformers.

**Strengths:**

- The paper is well motivated and clearly described for the most part. It provides sufficient background to make the reading self contained and explain what is the novel contribution of the work. It's also well situated and compared to prior work on ODE Transformers as it discusses prior first order and high-order methods with a single step.
- Builds on top of predictor-corrector methods from numerical analysis and extending them to improve their stability when training ODE Transformers. The key novelty lies in the selection of the predictor and corrector methods, and the proposal to use exponential moving average method to learn the coefficients in high-order methods instead of using constant values.
- The proposed method uses a high-order method as predictor and a multi-step method as corrector and aims to provide a better approximation to the implicit ODE problem in Transformers than previous studies.
- Presents empirical results on a variety of tasks are better than the original and previous ODE Transformers. This makes the work of interest to the researchers working in ODE Transformers.

**Weaknesses:**

- W1. There is lack of emphasis in the experiments on the computational overhead introduced by the high-order predictor and multi-step methods. Achieving better quality is not sufficient for adoption in practical settings. I'd suggest quantifying the training and inference cost compared to standard and ODE Transformers.
- W2. In terms of parameter efficiency, the proposed model achieves better performance with 1/3 of parameters only on one of the examined datasets. It's not clear if this is by chance or if the result holds on other settings. It would be useful to report performance with a smaller model size on the rest of the datasets to better establish the underlying claim.
- W3. The impact to research communities beyond the ones focusing on ODE transformers is somewhat limited because the results are with relatively small model sizes. Showcasing that the results hold on larger architectures such as 7B Mistral would make the results more convincing.
- W4. The improvement of the proposed multi-step high-order method compared alternative predictor-corrector methods is small (Table 10) and the benefit compared to simpler first-order methods from the predictor-only paradigm in a controlled setting is not provided (similar to the ablation in Table 10).

**Questions:**

- Q1. Where does the parameter efficiency of the model stems from and how a fair comparison with other models was ensured? In Appendix D, there is a section that discusses this but the comparison does not seem to be apples-to-apples (i.e. same config and #params) and whether the result holds in all the tasks in the experiment section.
- Q2. The method description wasn't entirely clear to me and I have the following questions:
   - The parameterization of $\mathcal{F}(P_{t+1}, \theta_t)$ is not clear from the textual description. What kind of transformation is used here?
   - How are the coefficients are parameterized exactly? Do you parameterize a single coefficient for each order for an $n$-order predictor or it is the same coefficient shared?
    - The text mentions that a larger weight is assigned ($a=0.5$) to the estimated $F_{t+1}$, is this value fixed or you have experimented with different values? It would be useful to clarify which coefficients among $\alpha$ and $\gamma$ are hyper-parameters and which ones are learned.
- Q3. Is the observed improvement worth the the additional computational cost for obtaining a better approximation? It would be useful to show what is the difference compared to a standard or a simpler ODE Transformer.
- Q4. Do you mean "ROUGE results" instead of "Rough results" in Table 4? In the same table, what is the number of parameters used by each model?
- Q5. What is the size of the models in Table 8 and is it the same for all variants?
- Q6. In the ablation experiment and other results, there is lack of comparison with a simple 1-order single/multi-step method or high-order 1-step method. Is the proposed method substantially better than them?
- Q7: Did you perform continued training of BERT directly on language understanding tasks or on some pre-training data first in Table 7? Please also report the number of parameters for both models and the exact configuration for PCformer.

**Limitations:**

The limitation section covers the shortcomings when the proposed methods are applied to different model designs but lacks discussion about the experiment shortcomings related to model size, computational cost, and comparison with simpler methods in a controlled setting.

---

> ### Author Rebuttal · Authors · 2024-08-06
>
> We thanks for your recognition for our motivation and the organization of our method.
> We also appreciate for your constructive feedback towards the shortcomes of the current manuscript, and we think all the concerns would be well addressed in our improved version. Here, we would like to show the details of your main concerns:
>
> > W1: Computational overhead: e.g., training and inference cost.
>
> We hope you can refer to the details of the general response W1. We report the training and inference cost of machine translation tasks and give a detailed analysis of large language models.
>
>
> > W2: Parameter efficiency: performance with a smaller model size on the rest of the datasets.
>
> - Besides OPUS, actually, we have already done simialr experiments in Figure 3 (Appendix), where a 28M PCformer can beat other models larger than it.
> - In our general response W2 for our newly proposed results on LLMs, we found that **Pcformer can beat the vanilla Transformer by only training on half of the training data! (1.3B models and 50B vs 100B tokens)**; More excited, our 340M PCformer (16B tokens) can achieve comparable results with 1.3B models with only 1/4 parameters!
> - For summarization tasks, a baseline with 12/18 encoder-layer configuration achieves RG-1, RG-2, and RG-L scores of 41.18, 18.23, 38.03 and 41.33, 18.31, 38.20, respectively. In comparison, our 2-order method yields scores of 41.96, 18.99, 38.74, **outperforming the baseline while utilizing only 1/3 to 1/2 of the layers.**
>
> We'll add more comprehensive comparisons next.
>
> > W3: Results on larger models such as Mistral 7B.
>
> - We appreciate your suggestion and completely agree. Our current results demonstrate that our model performs exceptionally well on smaller model sizes (less than 1B), indicating that PCformer has significant potential to scale up to larger model sizes. Given the interest from reviewers in seeing how PCformer performs in LLM evaluations, we have scaled up the model size from 340M to 1.3B parameters and increased the training data from 6B tokens to 100B tokens. These efforts have pushed the limits of our current computational resources. The summarized results are provided in the general response W2. We hope these results address your concerns.
>
> - We recognize the importance of showcasing results on even larger architectures like the 7B Mistral. However, achieving this would require additional computational resources. At present, we plan to train a PCformer based on the Mistral checkpoints and restart the training process with significantly less data compared to what was originally consumed in their pretraining. We are committed to conducting these more challenging experiments as soon as we have access to the necessary computational resources. Thank you for your valuable feedback.
>
>
> > W4: More details of the ablation (Table 10)
>
>
> Very good suggestions. Here we add the simpler first-order methods and ODE Transformer for each setting. The results are below:
>
> | #    | Predictor            | Corrector             | En-De | En-Fr | En-Ro | OPUS |
> | ---- | -------------------- | --------------------- | ----- | ----- | ----- | ---- |
> | 1    | First-order baseline | -                     | 29.91 | 43.22 | 34.20 | 31.5 |
> | 2    | ODE Transformer      | -                     | 30.77 | 43.96 | 35.28 | 32.3 |
> | 3    | RK2-block with EMA   | Multistep Method      | 30.70 | 44.27 | 35.55 | 33.7 |
> | 4    | RK2-block with EMA   | Backward Euler Method | 30.95 | 43.68 | 36.00 | 33.2 |
> | 5    | Multistep Method     | Multi-step Method     | 30.30 | 43.92 | 35.30 | 33.0 |
> | 6    | Multistep Method     | RK2-block with EMA    | 29.78 | 42.68 | 34.40 | 32.5 |
> | 7    | Multistep Method     | Backward Euler Method | 30.30 | 43.62 | 35.27 | 32.8 |
>
> We observe that all settings, except for the Multistep Predictor and High-order Corrector, show consistent BLEU improvements over the baseline (#1). This phenomenon is explained in our current manuscript: the predictor needs to be concise enough, otherwise, the initial value may cause the subsequent corrector computations to deviate. Additionally, both multistep and backward Eulere correctors show substantial improvements to ODE Transformer. Our proposed PCformer is a framework that you can choose a proper corrector according to the training date complexity, and RK2-block with EMA as the predictor always behaves well. We hope the results here can address your concern.
>
> > Q1: About fair comparison in experiments.
>
> We mainly want to compare the PCformer with the vanilla Transformer in two settings.
> - Firstly, as we only introduce quite small parameters, just a 1D tensor for EMA coefficient learning and several layernorm to achieve RK-Norm, we compare these two models in similar (near the same) parameters. The PCformer beats the vanilla Transformer by a quite large margin nearly in all scenarios.
> - Secondly, we can compare them in the same FLOPs, for example, a 6-layer PCformer with much fewer parameters can beat a 12-layer vanilla Transformer, meanwhile fewer computations.
> - Thirdly, for LLMs, we find that  PCformer also beats Transformer even the former trained with 50B tokens and the latter trained with 100B tokens. As we all know, increasing the training data is the most effective way and also the simplest way to improve performance. But at this serve setting, PCformer still behaves strongly.
>
> Due to the limited time and page, we will add more comparisons in all experiments for a much stronger claim.
>
> > Q2.1 The parameterization of  is not clear from the textual description. What kind of transformation is used here?
>
> As we emphasized that $\mathcal{F}_(P_{t+1},\theta_t)$ is the function to compute the derivate upon the input. Thus, either a self-attention network or an FFN, or even the whole encoder block could be regarded as an F function, also they could be seen functions in different granularities. In this work, for a fair comparison with ODE Transformer, we choose the whole block as the function.

---

> ### Author Response · Authors · 2024-08-06
> **Further response to the remained questions**
>
> > Q2.1 The parameterization is not clear from the textual description. What kind of transformation is used here?
>
> As we emphasized that $\mathcal{F}_(P_{t+1},\theta_t)$ is the function to computed the derivate upon the input. Thus, either a self-attention network or an FFN, or even the whole encoder block could be regarded as a F function, also they could be seen fuctions in different granularities. In this work, for a fair comparison with ODE Transformer, we choose the whole block as the function.
>
> > Q2.2 Details of $\alpha$
>
> Just a single coefficient. the code is as follows:
> ```
> self.alpha = torch.nn.Parameter(torch.Tensor(1))
> self.alpha.data.fill_(0.5)
> ```
> Take a 2-order EMA as an instance, the final layer ouput is computed as follows:
> ```
> x = residual + self.alpha*(1-self.alpha) * runge_kutta_list[0] + self.alpha*runge_kutta_list[1]
> ```
> where runge_kutta_list is a list to stores the previous obtained intermediate approximations ($\hat{F}_i$).
>
> We have uploaded our code to a anonymous github, more details could found in our codebase.
>
>
> > Q2.3 Whether $\alpha$ is fixed
>
> Similar with the previous question, we use 0.5 as an initail value for $\alpha$, while it is learnable. Thus $\alpha$ would be changed according to the graident decsent during the training phase. Sorry for the missing details for $\alpha$ and $\gamma$, and these two are both learnable tensors with an initial value 0.5. We will include this in our next version of the paper.
>
>
> > Q3. Is the observed improvement worth the the additional computational cost for obtaining a better approximation?
>
> As for the sequence generation models which follow a encoder-decoder paradigm,  our models are quite competive and the observed improvement is indeed worth the additional computational cost. Actually, we have already compared our PCformer with a simpler ODE Transformer in sequence generation tasks. We can see that, PCformer (RK2) can beat ODE Transformer (RK4) with  stronger performance and less computation cost. For example, in Table 1, PCformer (2-order) beats RK4-block (EMA) in both En-De and En-Fr tasks, with one less computation (The former 2 times predictor and 1 times corrector, while the later consumes 4 times forward computation for the high-order solution). SSimilar phenomena could be observed in abstractive summarization tasks (Table 2).
>
> > Q4: typos and the parameters of the models displayed in Table 4.
>
> Yes, we apologize for the typo; we mean "ROUGE results" instead of "Rough results" and will correct it in the next version. Regarding the number of parameters, all models listed in Table 4 have a similar parameter number, approximately 63M. This corresponds to a Transformer-base configuration, which includes 6 encoder layers and 6 decoder layers, each with a hidden size of 512 and 8 attention heads.
>
>
> > Q5: What is the size of the models in Table 8 and is it the same for all variants?
>
> The configurations of the models used to approximate the truncation errors are detailed in Appendix C.3. Specifically, we used a Transformer language model (decoder-only) with a hidden size of 512, tested in both 1-layer and 2-layer settings. Note that all all variants are the same. By comparing the results of the 1-layer and 2-layer models, we can see that both the ODE Transformer and our PCformer significantly reduce truncation errors, as measured by PPL. For instance, the PCformer (2nd order but with a 1-layer parameter) outperforms the Residual-Block (2-layer) and even surpasses the RK4-block (EMA) with fewer forward computations. This also provides context and a partial answer to your Q3 regarding computational efficiency.
>
> > Q6: Comparison with a simple 1-order single/multi-step method or high-order 1-step method.
>
> Yes, our PCformer significantly outperforms single/multi-step methods and high-order 1-step methods. Perhaps there was some oversight, but it is important to note that the single 1-step 1-order method corresponds to the vanilla Transformer (denoted as "Residual-block" in most tables). Transformer-DLCL represents a specific case of a multi-step method. We compared these models with our PCformer in Table 1. Additionally, the high-order 1-step method is represented by the ODE Transformer. We have already included comparisons with these methods in our MT experiments. We will ensure that these results are included in subsequent versions to provide a more comprehensive evaluation.
>
> > Q7: Details of the BERT training.
>
> Both the PCformer and the BERT models in Table 7 share the same parameter count, with approximately 335M parameters each. Specifically, both models have a hidden size of 1024, an FFN filter size of 4096, and 16 attention heads. We have aligned all settings with those specified in the original BERT paper to ensure a fair comparison. For PCformer, we pre-trained the model from scratch using a combination of WikiText and BookCorpus datasets. After the pre-training phase, we fine-tuned PCformer on the GLUE downstream tasks.

---

> > ### Comment · Reviewer_MB44 · 2024-08-12
> > **Response to authors**
> >
> > Thank you for the detailed responses to my questions and additional results! My main concerns have been addressed and I decided to increase my score:
> >
> > - The scaling experiments with LLMs are quite promising and increased my confidence that the results hold on larger model sizes.
> > - In terms of efficiency, the results show that there is no significant overhead introduced which addressed my main concern. It was also great to see results that justify the parameter efficiency (e.g. performance of PCFormer 340B vs Transformer 1.3B).
> > - The new ablation results clarify the differences compared to simpler baselines. It appears that the proposed methods have a fairly good improvement for translation that is greater than 0.5-1 point.
> >
> > The rest of the answers provided helpful clarifications, it would be great if they are reflected in the final version.

---

> > > ### Author Response · Authors · 2024-08-12
> > > **Thanks for the reconsideration**
> > >
> > > Thank you for reconsidering our work. We greatly appreciate the valuable suggestions you have provided, which we believe will further enhance our research. We plan to include these results and make thorough revisions in our next version. Thank you once again for your efforts!

---

### Official Review · Reviewer_Rc6M · 2024-07-12

**Soundness:** 3
**Presentation:** 3
**Contribution:** 3
**Rating:** 6
**Confidence:** 4

**Summary:**

This paper presents an approach to improve the performance of Transformer models for conditional natural language generation (machine translation and summarization).   The authors introduce a predictor-corrector framework, inspired by numerical methods for
solving ordinary differential equations (ODEs), to enhance the accuracy and
stability of the  models. The proposed model is evaluated against standard Transformer models on multiple benchmark datasets. The presented results show improvements in prediction accuracy and efficiency when compared to standard models.

**Strengths:**

The application of the predictor-corrector paradigm to Transformer models is innovative and offers a new perspective, extending the ODE transformer proposed in (Li et al., 31).

The integration of high-order ODE solutions into the Transformer architecture is a novel contribution. The authors use EMA (Exponential
Moving Average) coefficient learning to enhance training stability.

The experimental results look comprehensive, with evaluations on multiple benchmark datasets demonstrating the efficacy of the proposed model.

The paper provides a detailed description of the methodology.

The paper is well-organized and clearly written, with each section leading to the next.

**Weaknesses:**

The paper does not compare its results with state-of-the-art  models, which could have provided a more comprehensive evaluation of its effectiveness.  For instance for WMT14 En to FR, we can reach 43 in BLEU, instead of 41 for Attention is all you need.  Moreover, the difference with ODE Transformer is very limited.


The theoretical justification for the predictor-corrector framework could be more detailed, particularly in explaining how it relates to and improves upon existing methods.

The computational overhead induced by the proposed method is only lightly discussed in the Appendix and would require more in depth discussion.

**Questions:**

- How does the predictor-corrector framework perform in comparison to
state-of-the-art pretrained Transformer models?
- Do you think that all the 72 references are necessary?
- The quotation of Newell 59 is maybe a bit too much. Don't you think?

**Limitations:**

The authors should discuss the potential computational overhead introduced by the predictor-corrector framework.

---

> ### Author Rebuttal · Authors · 2024-08-06
>
> Thanks for your constructive advice and we think all the conerns would be well addressed in our improved version. We would like to address your concerns as follows:
>
> > W1: The paper does not compare its results with state-of-the-art models, which could have provided a more comprehensive evaluation of its effectiveness.
>
> Thank you for pointing out this issue. Due to page limitations, we focused our comparisons on the most closely related work, particularly those designed based on numerical methods, such as Transformer-DLCL, MacroNet, and ODE-Transformer. Notably, the ODE Transformer is a strong baseline on the WMT En-De and En-Fr benchmarks, used to achieving state-of-the-art results in these tasks. In addition, for the OPUS task, we compared PCformer with SoTA models like DeepNet and BranchNorm, which are leading models on this benchmark. To the best of our knowledge, PCformer indeed achieves SoTA results on WMT En-De (without pretrained models), En-Fr, En-Ro, and OPUS in these machine translation tasks. As for other tasks, such as summarization, PCformer shows competitive results compared to baselines learning from scratch. The current SoTA models in these tasks are primarily finetuned versions of BART and other advanced pretrained models. We will aim to make a fair comparison with these models in our next version. Thank you for your understanding and your valuable feedback.
>
>
> > W2: The theoretical justification for the predictor-corrector framework could be more detailed, particularly in explaining how it relates to and improves upon existing methods.
>
> - Compared to existing high-order methods such as Runge-Kutta (ODE Transformer), our PCformer leverages a predictor-corrector paramdigm to estimate intermediate approximations more accurately. Notably, we introduced an EMA-based coefficient learning method to enhance the robustness and stability of our high-order predictor, and the corrector then applies a multistep method to further reduce truncation errors.
>
> - Theoretical Improvements: Truncation error is a crucial factor in numerical solutions. In Section 3.1.2, we provide a theoretical analysis demonstrating that higher-order intermediate approximations tend to be more accurate, leading to more reliable final results. And in Table 8, we show that PCformer can achieve lower ppl (analogous to truncation errors) than the 1-step and high-order methods.
>
> - Practical Superiority: As demonstrated in our experimental results section, our framework significantly outperforms existing models, including the robust 3.8B DeepNet, on several large-scale datasets while using only one-third of the parameters.
>
>
> > W3: The computational overhead induced by the proposed method is only lightly discussed in the Appendix and would require more in depth discussion.
>
> Thank you for your advice. We regret that we were unable to include this discussion in the main body of this paper due to time constraints. We would like to provide the following summary on the complexity and computational overhead induced by our proposed method, the details please refer to the general response W1!
>
> In a nutshell, PCformer only introduces slight inference overhead as we mainly applied the predictor-corrector in the encoder side, which takes a very small amount of computation in the process of auto-regressive decoding.
>
>
> > Q1: How does the predictor-corrector framework perform in comparison to state-of-the-art pretrained Transformer models?
>
> We have conducted extensive experiments across several benchmarks. Our proposed PCformer not only significantly outperforms the vanilla Transformer but also shows a substantial improvement over the strong ODE Transformer. We believe your question may be directed towards understanding how PCformer performs in comparison to other large pretrained language models (LLMs). If our interpretation is incorrect, please let us know, and we will address the specifics accordingly.
>
> Due to the limited number of response characters, please refer to the general response W2 for the new proposed results on LM harness evaluation. We can see that PCformer beat the Transformer baseline across all settings, we will further scale the model to a much larger capactiy in the future, e.g. 7B/8B model to make a fair comparison with LLAMA models.  While training even larger LMs from scratch is costly and time-consuming, it is a promising direction for further research.
>
> > Q2: Do you think that all the 72 references are necessary?
>
> We have carefully reviewed all the references cited in our work and believe that almost all reference are necessary. The large number of references is due to the comprehensive nature of our study, which spans five distinct tasks: machine translation, abstractive summarization, language modeling, natural language understanding, and LM harness evaluation. For each task, we report results on several widely used benchmarks, such as WMT En-De, En-Fr, En-Ro, OPUS, nine test sets for BERT, and eight test sets for LM harness evaluation. To ensure a fair comparison and to validate our results, it was essential to cite the datasets and the results from related work. Consequently, the extensive referencing reflects the breadth and depth of our study rather than an excess. We will futher check the details to avoid unnecessary reference included!
>
> > Q3: The quotation of Newell 59 is maybe a bit too much. Don't you think?
>
> Thank you for pointing this out. Our intention was to illustrate that the predictor-corrector paradigm aligns with the seminal ideas of Newell, as also mentioned in the Tree-of-Thought paper. Specifically, the predictor first provides an initial estimate. This estimate is then refined by the corrector for greater accuracy, mirroring the problem-solving process proposed by Newell. We appreciate your feedback and will carefully reconsider and reorganize this section in the next version of our manuscript to ensure it is succinct and relevant.

---

> ### Comment · Reviewer_Rc6M · 2024-08-13
>
> Thank you for all your comments and discussions. I acknowledge I have read them, but I stand by my original review.

---

### Official Review · Reviewer_1pKC · 2024-07-14

**Soundness:** 3
**Presentation:** 3
**Contribution:** 2
**Rating:** 5
**Confidence:** 1

**Summary:**

The paper presents advancements in Transformer architecture to minimize errors in approximating solutions to Ordinary Differential Equations (ODEs).

The contributions are:
- Introducing a learning paradigm with a high-order predictor and multistep corrector to reduce truncation errors.
- Proposing an exponential moving average-based method to enhance the predictor's learning ability and stability by replacing constant coefficients with dynamic ones.
- Demonstrating superior performance across various benchmarks, including machine translation, abstractive summarization, language modeling, and natural language understanding, achieving notable improvements in BLEU scores and parameter efficiency.

The work shows improvements in translation tasks and highlights the general applicability of the proposed methods across different natural language processing domains.

**Strengths:**

The paper introduces a new predictor-corrector framework within Transformer architectures, which is a fresh and innovative approach to addressing errors in approximating solutions to ODEs. The integration of a high-order predictor and multistep corrector, combined with an exponential moving average (EMA) coefficient learning method, represents a creative combination of established numerical analysis techniques with modern neural network architectures.

The paper successfully applies the method across various natural language processing tasks. The paper is well-structured and clearly written, making complex concepts accessible.

**Weaknesses:**

The complexity of implementing these methods might be a barrier for practical adoption. The paper could benefit from providing more detailed guidelines or code to facilitate easier implementation and replication of the results.

The experiments primarily focus on natural language processing tasks. While the results are impressive, it remains unclear how well the proposed methods generalize to other domains, such as time series forecasting. Including preliminary results or discussions on the potential applicability to these other areas could strengthen the paper.

**Questions:**

Can you provide more details on the computational cost of the proposed predictor-corrector framework and EMA coefficient learning method? Specifically, how do these methods impact training time and resource utilization compared to traditional Transformer models?

How does the proposed method scale with increasing model size and dataset complexity? Have you encountered any challenges in scaling up your approach, and if so, how did you address them?

**Limitations:**

Provide a detailed analysis of the computational complexity and resource requirements of the proposed methods. This should include a comparison with standard Transformer models and an explanation of any trade-offs between performance improvements and computational costs.

Discuss the scalability of the proposed methods in more detail. Explain any challenges encountered when scaling up to larger datasets or models and how these were addressed. Provide insights into potential limitations in terms of scalability and how future work could overcome these challenges.

---

> ### Author Rebuttal · Authors · 2024-08-06
>
> Thanks for your recognition on our writting and the style to present our idea. We would like to answer your questions as follows:
>
> > W1: The complexity of implementing these methods might be a barrier for practical adoption. The paper could benefit from providing more detailed guidelines or code to facilitate easier implementation and replication of the results.
>
> We open-source our core code at https://anonymous.4open.science/r/Neurips-PCformer/. This allows others to easily reproduce our results and apply PCformer to additional benchmarks. We will also release the complete codebase after finalizing the README.md to facilitate better reproducibility.
>
>
> > W2: The experiments primarily focus on natural language processing tasks. While the results are impressive, it remains unclear how well the proposed methods generalize to other domains, such as time series forecasting. Including preliminary results or discussions on the potential applicability to these other areas could strengthen the paper.
>
> We mainly evaluate our PCformer on NLP tasks, including both the natural language understanding and natural language generation. While, our method is quite general that it could be applied to other Transformer variants in other areas, e.g., Swin-Transformer in computer vision. For example, our PCformer can also improve Swin-Transformer in tiny and base configurations. The results were conducted on 224*224 image size, and 2-order predictor with a multistep corrector.
>
>
> | Model                  | Params. | Image size | Top1-acc |
> | ---------------------- | ------- | ---------- | -------- |
> | Swin-Transformer(tiny) | 29M     | 224*224    | 81.3     |
> | PCformer(tiny)         | 29M     | 224*224    | 82.0     |
> | Swin-Transformer(base) | 88M     | 224*224    | 83.5     |
> | PCformer(base)         | 88M     | 224*224    | 84.0     |
>
>
> Beside this, we also follow your suggestion to evaluate PCformer on time-series forecasting tasks. We select 10 multivariate datasets from UEA Time Series Classification Archive following the setting and the codebase provided by Flowformer [1]. Thus we choose the Flowformer as the baseline, which is also a strong model on these testsets. For the details, we build the PCformer upon Flowformer and report the 2-order predictor and Euler corrector as the training data is very small. Also, we use RK-Norm to avoid the model suffering from the overfitting problem as the authors of Flowformer trained their models upon to 100 epochs (or even 400 epoch on some tasks.). The results are evaluated by the best accuracy. We can see that PCformer can beat the Flowformer by 2 average score on 10 testsets, which demonstrates the effectiveness on time-series forecasting tasks. We hope these results can address your concern, and we'd like to add them into the updated version of this paper.
>
>
> | Dataset              | Flowformer | PCformer |
> | -------------------- | ---------- | -------- |
> | EthanolConcentration | 30.3       | 33.9     |
> | FaceDetection        | 67.0       | 68.2     |
> | HandWriting          | 29.1       | 33.5     |
> | Heartheat            | 77.0       | 78.5     |
> | JapaneseVowels       | 98.4       | 99.2     |
> | PEMS-SF              | 87.2       | 87.9     |
> | SelfRegulationSCP1   | 89.0       | 92.2     |
> | SelfRegulationSCP2   | 55.0       | 56.1     |
> | SpokenARABICDIGITS   | 98.0       | 100.0    |
> | UWAVEGESTURELIBRARY  | 85.3       | 86.3     |
> | Average Score        | 71.6       | 73.6     |
>
> [1] Flowformer: Linearizing Transformers with Conservation Flows, ICML 2022.
>
> > Q1: Can you provide more details on the computational cost of the proposed predictor-corrector framework and EMA coefficient learning method? Specifically, how do these methods impact training time and resource utilization compared to traditional Transformer models?
>
> Our proposed PCformer is parameter-efficient as we share the parameters between the predictor and the corrector, especially for the F functions in the high-order predictor. Please refer to the general response W1 for more details.
>
>
> > Q2: How does the proposed method scale with increasing model size and dataset complexity? Have you encountered any challenges in scaling up your approach, and if so, how did you address them?
>
> Our PCformer could be easily scaled from both the model capacity and training tokens. We summarize the results of larger model capacity (from 340M to 1.3B) and training tokens (from 6B to 100B) in the general response W2, where PCformer can beat LLama-like models in all scenarios. Additionally, the results in Table 3 have already shown the superiority of PCformer in a 1.2B setting on OPUS multilingual translation tasks.
>
>  We think there is only a major challenge: the computational overhead of PCformer is somewhat higher than that of the vanilla Transformer (e.g., LLama models). This limitation has been discussed in our paper. We are actively working to address the training and inference overhead in our ongoing research.

---

> ### Author Response · Authors · 2024-08-12
> **Any Further Concerns About Our Work**
>
> Dear Reviewer 1pKC,
>
> We apologize for reaching out as the discussion deadline approaches. We are grateful for the comprehensive and useful feedback you provided, and we have responded in detail to your comments during the rebuttal phase, e.g., new results on time-series forecasting, the practical cost of training and inference, more strong results on larger LLMs using PCformer and so on.
>
> We are eager to know if our proposed results and clarifications have adequately addressed your concerns. If so, we would appreciate it if you could reconsider your score.
>
> Thank you once again for your time and effort.
>
> Best regards,
> The Authors

---

> > ### Author Response · Authors · 2024-08-13
> > **Looking Forward to follow-up discussion**
> >
> > Dear Reviewer 1pKC,
> >
> > Thank you once again for your suggestions and valuable feedback. We apologize for reaching out again as the deadline approaches. To address your concerns, we have conducted experiments on both the time-series forecasting task and the image classification task. Additionally, we have provided data on the practical costs of training and inference for both the encoder-decoder and decoder-only paradigms. We have now uploaded the results of a 3B PCformer using 16B and 50B tokens, respectively. The results show that there are no major challenges for its scaling up. If our results and explanations help address your concerns, we would be grateful if you could acknowledge our rebuttal and consider adjusting your score accordingly.
> >
> > Best wishes
> >
> > The Authors.

---

### Author Rebuttal · Authors · 2024-08-06

Thank you to all four reviewers for your efforts and instructive comments on our paper. We believe these updated results address your concerns regarding the efficiency and effectiveness.

> W1: about the computation overhead, inference and training cost comparison.

| Model                    | Layers | Inference | Memory | BLEU |
| ------------------------ | ------ | --------- | ------ | ---- |
| Transformer              | 6      | 98.7      | 13.2   | 29.2 |
| Transformer              | 12     | 94.5      | 18.7   | 29.7 |
| Transformer              | 24     | 87.3      | 23.5   | 29.8 |
| ODE Transformer (RK2)    | 6      | 93.5      | 15.1   | 30.7 |
| PCformer (RK2 predcitor) | 6      | 90.3      | 16.2   | 30.9 |
| ODE Transformer (RK4)    | 6      | 87.1      | 17.3   | 30.5 |


1. The table above compares the inference speed (sentences/s) and memory consumption (GB) for various models in a big configuration on WMT En-De. The experimental results show that the proposed PCformer models achieve comparable inference speeds with the baselines. This is primarily because MT models typically follow an encoder-decoder paradigm, with the main computational overhead coming from the decoder side due to the auto-regressive decoding process, rather than the encoder on which we primarily experimented.

2. In terms of memory consumption, PCformer is also competitive. It consumes slightly more memory during the forward computation phase because we need to store previously obtained approximations for the multistep method (corrector), as well as iteratively generated inner step approximations for higher-order predictors. This slightly increases the memory consumption for encoder-decoder models.

3. However, if the proposed predictor-corrector paradigm is applied to encoder-only or decoder-only models, such as BERT for the former and LLMs for the latter, the additional overhead is nonnegligible. For example, our PCformer would consume about twice time compared the vanilla baseline in LLM training. Despite this, the performance gains are still noticeable. These observations motivate us to develop more efficient variants to overcome the computational overhead.

4. Fortunately, we have already collected some promising results in this direction. We attempted to achieve the high-order approximations in latent space and current experimental results show there is only a quite small performance gap compared with the PCformer in most of tasks, but is computation friendly!


W2: More results on LLMs.

| Model(Params & Tokens)     | Wiki.(ppl) | LMB.(ppl) | LMB.     | PIQA     | Hella.   | SCIQ     | ARC-c    | Winograde| Avg.     |
| -------------------------- | ---------- | --------- | -------- | -------- | -------- | -------- | -------- | -------- | -------- |
| Transformer++ (340M & 6B)  | 38.5       | 96.1      | 21.4     | 60.3     | 29.1     | 69.2     | 21.5     | 50.4     | 41.9     |
| PCformer (340M & 6B)       | 35.3       | 78.8      | 23.6     | 61.6     | 30.1     | 71.6     | 22.9     | 51.8     | 43.6     |
| Transformer++ (340M & 16B) | 28.3       | 65.3      | 29.8     | 63.2     | 33.9     | 73.2     | 23.1     | 51.4     | 45.8     |
| PCformer (340M & 16B)      | 25.6       | 39.7      | 34.5     | 65.2     | 36.9     | 79.6     | 23.2     | 52.2     | 48.6     |
| Transformer++ (1.3B & 16B) | 23.8       | 26.2      | 37.3     | 65.7     | 37.6     | 78.6     | 23.7     | 51.5     | 49.0     |
| PCformer (1.3B & 16B)      | 20.9       | 23.2      | 42.5     | 68.3     | 43.4     | 81.5     | 25.1     | 52.4     | 52.2     |
| Transformer++(1.3B & 100B) | 16.3       | 11.8      | 51.6     | 71.0     | 51.7     | 86.7     | 28.1     | 54.6     | 57.2     |
| PCformer (1.3B & 50B)      | 16.2       | 9.38      | 55.1     | 71.9     | 54.8     | 88.6     | 29.6     | 57.2     | 59.5     |
| PCformer (1.3B & 100B)     | **14.0**   | **7.46**  | **59.6** | **73.8** | **60.0** | **90.7** | **31.7** | **61.7** | **62.9** |

1. Here we scaled the model size from 340M to 1.3B parameters, which is the maximum size feasible within the limited time. For the data, we scaled from 6B to 16B and 100B tokens. PCformer can significantly beats the baseline in similar parameters, and with more training data, the average performance improves significantly. This demonstrates that our model benefits from increased data size, without experiencing diminishing returns. This indicates that model bias persists and plays a crucial role in our setting (Larger settings still worth to be explored in future work). As the model size increases (from 340M to 1.3B), PCformer shows substantial performance gains, both in terms of accuracy on the LM harness evaluation and lower perplexity on Wikitext and Lambada tests.

2. We trained the 1.3B model on a cluster of 256 A100 GPUs. Note that 1.3B models consist of 24 layers, where the hidden size is 2048 and the FFN size is 5432 （8/3 * hidden size, 16 attention heads and SiLU activation functions. The baseline (1.3B + 100B tokens) was trained up to 20 hours and nearly 40 hours for our PCformer. Thus PCformer (1.3B + 50B tokens) was trained within similar time.

3. Additionally, given that PCformer consumes more FLOPS per forward pass, we compared models with similar total FLOPS consumption.
    - Our PCformer trained on less than 50B data outperformed Transformer++ trained on 100B data by a significant margin. Additionally, PCformer (340M & 16B) achieved results comparable to Transformer (1.3B & 16B) with nearly 1/4 of the parameters. These results demonstrate that PCformer remains competitive in settings with larger model capacity and data volumes, highlighting the potential for further research in model designs that fully utilize available data.
    - While the increased inference and training costs in the decoder-only paradigm are notable, the substantial performance gains justify continued exploration of PCformer, including parameter-efficient training algorithms.

---

### Author Response · Authors · 2024-08-10
**Any Other Concerns on PCformer**

We extend our heartfelt gratitude to the four reviewers for their insightful feedback and to the Area Chairs for facilitating the rebuttal discussion. We appreciate the time and effort you have invested in evaluating our submission.

Since uploading our rebuttal, we have not yet received any further feedback. We eagerly look forward to engaging in discussions with the reviewers and believe that we have effectively addressed the major concerns raised. We also welcome any other further concerns!

We are aware that the ACL conference is approaching, and we hope you all have wonderful days

---

### Author Response · Authors · 2024-08-13
**Further Results on a 3B PCformer**

| Model(Params & Tokens)     | Wiki.(ppl) | LMB.(ppl) | LMB.     | PIQA     | Hella.   | SCIQ     | ARC-c    | Winograde| Avg.     |
| -------------------------- | ---------- | --------- | -------- | -------- | -------- | -------- | -------- | -------- | -------- |
| Transformer++ (340M & 6B)  | 38.5       | 96.1      | 21.4     | 60.3     | 29.1     | 69.2     | 21.5     | 50.4     | 41.9     |
| PCformer (340M & 6B)       | 35.3       | 78.8      | 23.6     | 61.6     | 30.1     | 71.6     | 22.9     | 51.8     | 43.6     |
| Transformer++ (340M & 16B) | 28.3       | 65.3      | 29.8     | 63.2     | 33.9     | 73.2     | 23.1     | 51.4     | 45.8     |
| PCformer (340M & 16B)      | 25.6       | 39.7      | 34.5     | 65.2     | 36.9     | 79.6     | 23.2     | 52.2     | 48.6     |
| Transformer++ (1.3B & 16B) | 23.8       | 26.2      | 37.3     | 65.7     | 37.6     | 78.6     | 23.7     | 51.5     | 49.0     |
| PCformer (1.3B & 16B)      | 20.9       | 23.2      | 42.5     | 68.3     | 43.4     | 81.5     | 25.1     | 52.4     | 52.2     |
| Transformer++(1.3B & 100B) | 16.3       | 11.8      | 51.6     | 71.0     | 51.7     | 86.7     | 28.1     | 54.6     | 57.2     |
| PCformer (1.3B & 50B)      | 16.2       | 9.38      | 55.1     | 71.9     | 54.8     | 88.6     | 29.6     | 57.2     | 59.5     |
| PCformer (1.3B & 100B)     | **14.0**   | **7.46**  | **59.6** | **73.8** | **60.0** | **90.7** | **31.7** | **61.7** | **62.9** |
| PCformer (3B & 16B)        | 17.8       | 13.6      | 48.0     | 69.7     | 48.4     | 83.1     | 26.4     | 55.2     | 55.1     |
| PCformer (3B & 50B)        | **13.6**   | **6.5**   | **62.1** | **74.4** | **61.9** | **90.6** | **32.4** | **61.9** | **63.9** |



Since the discussion deadline is approaching, we would like to once again thank all the reviewers for their efforts in helping to further improve our work. Given the limited time, we have focused on reporting the results of PCformer on a 1.3B model using 100B tokens in the rebuttal phase. Additionally, we have attempted to further scale the model to over 3B parameters, with a hidden size of 3200, an intermediate size of 8640, and 26 layers, following the LLama3B configuration. We trained this model using 128 A100 GPUs and, up to now, it has been trained on nearly 50B tokens. Below are the results for the 3B model at different stages of training (16B tokens and 50B tokens):

- PCformer (3B & 16B) outperforms its 1.3B & 16B counterpart by an average score of 2.9, and PCformer (3B & 50B) surpasses PCformer (1.3B & 50B) by an average score of 4.4. This indicates that with increased data consumption, PCformer continues to benefit from the data and demonstrates superior results.

- PCformer (3B & 50B) achieves an average score 8.8 points higher than PCformer (3B & 16B), and PCformer (1.3B & 50B) exceeds PCformer (1.3B & 16B) by an average score of 7.3. This suggests that the performance gap widens with increasing data volumes, which is intuitive as larger models naturally require more data for effective training.

The above two phenomena demonstrate the scalability of PCformer, and we are enthusiastic about the opportunity to demonstrate its potential to the community. Thank you once again to all the reviewers, ACs, and SACs for your invaluable feedback and support.

---

### Decision · Program_Chairs · 2024-09-25

**Decision:**

Accept (poster)

**Comment:**

This paper explores transformer architecture design to minimize the error compared to the solutions from ODE formulation. It introduces a predictor-corrector learning framework to minimize the truncation errors. It also proposes an exponential moving average-based coefficient learning method to further strengthen the higher-order predictor. Extensive experiments show the superiority of the approach on different benchmarks. All the reviewers acknowledge that this paper provides novel contribution and has valuable contribution to transformer architecture innovation from ODE perspective. However, the reviewers have also pointed out several weakness of the paper, such as restrictions of the application to natural language domain, relatively small model size in experiments, and potential complexity of the method. During the rebuttal, the authors have successfully addressed most of the major concerns by providing additional results on vision domain and time series domain. It also provides additional experiments on the scaling behavior of the method, which shows its potential effectiveness when the FLOPs go up. One suggestion from my side to present the scaling law results is to also show the results in the form of “performance vs FLOPs”, which is another standard visualization in scaling law study.